# An approach for long-term, multi-probe Neuropixels recordings in unrestrained rats

Thomas Zhihao Luo[1†*], Adrian Gopnik Bondy[1†*], Diksha Gupta[1], Verity Alexander Elliott[1], Charles D Kopec[1], Carlos D Brody[1,2]

[1]Princeton Neuroscience Institute, Princeton, United States; [2]Howard Hughes Medical Institute, Princeton University, Princeton, United States

**Abstract** The use of Neuropixels probes for chronic neural recordings is in its infancy and initial studies leave questions about long-term stability and probe reusability unaddressed. Here, we demonstrate a new approach for chronic Neuropixels recordings over a period of months in freely moving rats. Our approach allows multiple probes per rat and multiple cycles of probe reuse. We found that hundreds of units could be recorded for multiple months, but that yields depended systematically on anatomical position. Explanted probes displayed a small increase in noise compared to unimplanted probes, but this was insufficient to impair future single-unit recordings. We conclude that cost-effective, multi-region, and multi-probe Neuropixels recordings can be carried out with high yields over multiple months in rats or other similarly sized animals. Our methods and observations may facilitate the standardization of chronic recording from Neuropixels probes in freely moving animals.

*For correspondence:
thomas.zhihao.luo@gmail.com (TZL);
adrian.bondy@gmail.com (AGB)

†These authors contributed equally to this work

Competing interests: The authors declare that no competing interests exist.

## Introduction

Behavior depends on the concurrent spiking activity of many neurons across large distances in the brain. Therefore, tools that improve our ability to sample neuronal activity at the temporal resolution of action potentials and also with large spatial coverage will expand our ability to answer key neuroscience questions. The recent development of a silicon microelectrode array, the Neuropixels probe (*Jun et al., 2017*; *Mora Lopez et al., 2017*), is a significant step forward in that direction. Its high channel count allows sampling from hundreds of neurons per probe, on-probe amplification and digitization allows for low-noise data transmission, and a small footprint allows multiple probes to simultaneously survey regions across the brain. Given these advantages, the use of Neuropixels probes in acute experiments in head-fixed mice has become widespread and standardized (*Allen et al., 2019*; *Steinmetz et al., 2019*; *Aguillon-Rodriguez et al., 2020*).

Chronic implantations are particularly crucial for experiments that examine behaviors involving unrestrained movement, such as some perceptual decision-making (*Brunton et al., 2013*) tasks, exploration (*Tervo et al., 2014*), navigation (*Ito et al., 2015*), escape (*Juavinett et al., 2019*), foraging (*Davidson and El Hady, 2019*), and prey-capture (*Hoy et al., 2019*). However, the deployment of these probes in experiments requiring chronic implantation has remained limited (*Juavinett et al., 2019*; *Jun et al., 2017*; *Krupic et al., 2018*). A previous study tracked the stability of neural signals acquired from the rat medial prefrontal cortex (*Jun et al., 2017*). This study found that across implants, there was no significant degradation of signals across 2 months. A different study reported that 20–145 single units could be isolated in the mouse visual cortex, hippocampus, and midbrain from 1 to 2 weeks after surgery (*Juavinett et al., 2019*). This latter study used an implantation strategy that allowed the probe to be recovered and reused, which is highly

advantageous given the significant cost of the probes. Moreover, this study reported that high-quality neural signals could be observed after re-implantation of two explanted probes.

Several outstanding issues need to be addressed before the enormous potential for chronic implantation of Neuropixels probes can be fully realized. First, the long-term yield across different brain regions is unknown because prior chronic studies recorded from a small set of brain regions using test-phase probes (*Juavinett et al., 2019*; *Jun et al., 2017*). The stability of spiking signals is likely to vary across regions because the brain is not mechanically uniform, with different viscoelastic properties and levels of respiration- and cardiac-related movement in different regions (*Bayly et al., 2005*; *Budday et al., 2015*; *MacManus et al., 2018*; *Sloots et al., 2020*). Direct measurements comparing the long-term stability of Neuropixels recordings across the brain are needed to determine when it is advantageous to use these probes. Second, the feasibility of probe reuse remains uncertain. The reusability of explanted probes was assessed previously by comparing the frequency and signal-to-noise of spiking events recorded from two re-implanted probes (*Juavinett et al., 2019*). However, neural signals can vary across implantations for reasons unrelated to the probes themselves. To assess the feasibility of probe re-use, it is necessary to also measure the performance of the explanted probe independent of neural signals. Moreover, techniques that allow probe recovery have not been validated for multi-month recordings, or validated in rats, which can be ten times larger in size than mice and can generate much larger impact forces that can damage probe signals.

To address these issues, we developed and validated an approach for recoverable chronic implants that is robust to forces generated by rats over multiple months and compact enough for multiple probes to be implanted on the same rat. We designed a probe holder, inspired by *Juavinett et al., 2019*, that can be produced using a 3D printer and allows the probe to be easily retrieved for reuse. Using this approach, we recorded from 18 rats, each implanted with up to three probes simultaneously, while they performed a cognitively demanding task for multiple hours daily without human supervision and at a level similar to their pre-implantation baseline. Recordings were made broadly throughout the rat forebrain and midbrain. From these, we reached three conclusions: first, over a hundred single units can be recorded per probe over multiple months; second, the stability of long-term recording depends systematically on the dorsoventral and anteroposterior position in the brain, with greater stability for more anterior and ventral regions; and third, to test the feasibility of re-using Neuropixels probes after extended periods of implantation, we directly measured the input-referred noise of explanted probes in saline after prolonged implantation. We found only modest increases in noise across multiple (up to three) cycles of implantation and explantation, demonstrating the practicality of probe re-use. Taken together, these results further demonstrate the enormous potential of Neuropixels probes for experiments requiring unrestrained movement, validate new methods for deploying these probes to full advantage in chronic preparations, and provide data on real-world performance that can guide chronic experiments and future probe development.

## Results

### Design overview

The principal element of the system is a highly compact implant assembly (*Figure 1A–C*) for mounting and enclosing individual Neuropixels probes. The maximum dimensions of the assembly are 43 mm (height), 25 mm (width), and 10.5 mm (depth), with a weight of 2.6 g. The size and weight of the design provides sufficient protection to the probe from the impact force that a rat can generate while allowing for a rat to comfortably carry additional probes or hardware for neural perturbation. We minimized the cross-section of the assembly as much as possible towards the side facing the brain, so that multiple such assemblies could be implanted on the same animal. By inserting each probe at an angle, a large range of configurations can be achieved (see photos of multi-probe implants in *Figure 1F* and Figure 3C).

The implant assembly contains four discrete parts, all of which we printed in-house using Formlabs stereolithography 3D printers. The first and smallest part is a *dovetail adapter* (*Figure 1A*). On one side, it consists of a flat platform that was permanently glued to the probe base. One the other side it contains dovetail rails to allow it to mate with the rest of the assembly. The second part is an *internal holder* (*Figure 1B*), whose primary purpose is to allow the probe to be stereotaxically

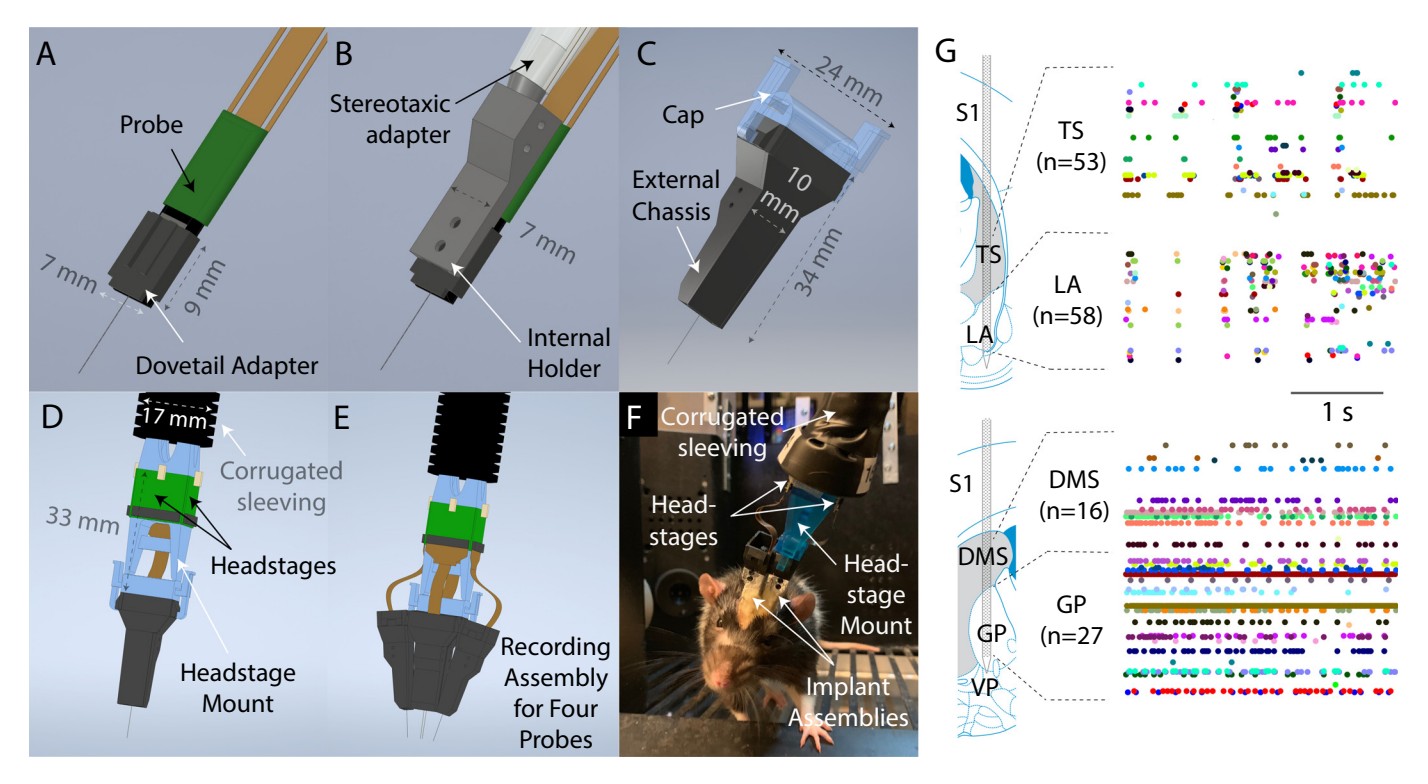

**Figure 1.** Design overview. (**A**) To prepare a probe for implantation, it is first mounted to a dovetail adapter. (**B**) The dovetail adapter mates with an internal holder through a dovetail joint. The internal holder can be manipulated with a commercially available stereotaxic holder. (**C**) The entire apparatus is then enclosed with, and screwed onto, an external chassis. During the surgery, the external chassis, but not the internal holder, is fixed to the skull surface. Therefore, the internal holder and probe could be fully removed at the end of the experiment simply by unscrewing them from the external chassis. The external chassis contains a mechanism to allow a cap to be latched on top to protect the probe when recording is not taking place. (**D**) Using the same latching mechanism, a headstage mount can be connected to the assembly for recording instead of the cap shown in (**C**). The headstage mount is fixed to a corrugated plastic sleeving, which protects the cables from small bending radii, and contains space for up to four headstages. (**E**) Schematic illustration of the full recording system for four probes simultaneously implanted. (**F**) A photograph of a rat in a behavioral rig with two probes implanted and with a headstage mount connected for recording. (**G**) Simultaneous recordings from two Neuropixels probes in a moving rat. Top: recording from tail of striatum (TS) and lateral amygdala (LA). Bottom: recording from dorsomedial striatum (DMS) and globus pallidus (GP). Images and schematic are adapted from *Paxinos and Watson, 2006*. Colored dots indicate spike times.

manipulated. It mates with the dovetail rails on the dovetail adapter and contains a cylindrical protrusion on its dorsal face compatible with a stereotaxic cannula holder. The third and fourth element jointly make up the *external chassis* (*Figure 1C*), which fully encase the other components. This provides mechanical protection and prevents adhesive materials applied during surgery from contacting the other components. The external chassis is printed in two parts which are glued together on either side of the internal holder during construction of the implant. During implantation, only the external chassis is cemented to the animal's skull. At the end of an experiment, the screws attaching the external chassis to the internal holder can be removed to allow the probe and internal holder to be explanted simply by lifting them away from the implant. Furthermore, by removing the dovetail adapter from the internal holder, a nearly bare probe can then be recovered.

Instead of devoting space in the implant assembly for a headstage mount, as a previous design did (*Juavinett et al., 2019*), we opted to provide the external chassis with a latching mechanism so that additional hardware can be attached to the implant as needed. When recording is not taking place, the probe's flex cable is coiled inside the external chassis and a 3D-printed cap (*Figure 1C*) is latched in place to provide protection for the probe. During recording, the cap is removed and a headstage mount is attached in its place (*Figure 1D–F*). Up to four headstages can be mounted on

the headstage mount, allowing up to four probes to be recorded simultaneously (*Figure 1E*). The headstage mount is taped to a tether made of ½"-diameter corrugated plastic sleeving, which protects the delicate Neuropixels cable from small bending radii (*Figure 1F*).

Using this system, 18 rats were implanted with one or more Neuropixels probes. The probes used were commercially available Neuropixels 1.0 probes (IMEC, Leuven, Belgium), with the option of a flat silicon spacer attached parallel to the plane of the probe, except for one (among 20), which was phase 3A option one probe. One rat was implanted with three probes simultaneously, and two rats with two concurrent probes (*Figure 1F–G*). The probes targeted a large number of cortical and subcortical brain regions, including prefrontal, visual, somatosensory, and retrosplenial cortex, as well as the basal ganglia, amygdala, and the superior and inferior colliculi. See *Figure 4—figure supplement 1* for example histological images, *Table 1* for a list of each individual implantation and its outcome, and *Table 2* for each brain region targeted by each implantation.

Constructing the implant assembly, and mounting a probe to it, typically requires no more than 1 hr, once all parts are printed and required tools are on hand. A detailed protocol is available in the Materials and methods, along with illustrations of each step in *Figure 2*. A detailed protocol for implantation and explantation of probes using our system is also in the Materials and methods, with illustrations of key steps in *Figure 3*.

## Stability of spiking signals

To quantify the stability of spiking signals across days, we periodically performed short recordings of approximately ten minutes from the group of 384 recording sites either closest or second closest to the tip of the probe shank ('bank 0' and 'bank 1', respectively). A total of 46,929 isolated units were recorded across 141 recording sessions, 18 animals, and 15 unique stereotaxic coordinates. Units recorded on the same probe but on different days were considered independent. All but one probe implanted were Neuropixels 1.0 probes, and all results were identical if the testing-phase ('3A') probe was excluded.

For spike sorting, we used Kilosort2 (*Pachitariu, 2020*). Isolated units that exhibited sufficiently few refractory period violations were classified as 'single units' (SUs) and the rest as 'multi-units' (MUs) (Materials and methods). We made the explicit choice to use the default parameters and report results without subsequent manual curation. This has the advantage of being entirely reproducible and immune from the subjectivity of human involvement (*Buccino et al., 2020*; *Harris et al., 2000*; *Wood et al., 2004*). Nonetheless, we did perform manual curation on a subset of the data (n = 25 sessions), and a comparison of yields before and after manual curation is shown in *Figure 4— figure supplement 2*. Manual curation primarily had the effect of uniformly scaling down the number of identified units (correlation between pre- and post-curation yields across sessions was >0.9; *Figure 4—figure supplement 3*, panels E,F), and therefore is likely to have only a small effect on the differences in yield between sessions, which was of primary interest here (*Figure 4—figure supplement 2I,J*). Similarly, we would expect other spike sorters to primarily affect the absolute number of units detected. The comparative performance of different spike sorters is an area of active research (*Buccino et al., 2020*) that we did not explore here.

The brain areas where we recorded include the medial prefrontal cortex (*Figure 4A*), dorsomedial frontal cortex, motor cortex (*Figure 4B*), nucleus accumbens (*Figure 4C*), dorsomedial striatum, piriform cortex, somatosensory cortex, globus pallidus, ventral pallidum, amygdaloid complex, striatum tail, dorsal-posterior cortex, the postsubiculum, the superior and inferior colliculi, and posterior tectum. The SU yield of each implant separated by brain area is shown in *Figure 4—figure supplement 4*, and the details of the brain areas targeted by each implant are provided in *Table 2*.

Averaged across implanted probes, the total number of isolated units (including SUs and MUs) per electrode began at 1.51 (bootstrapped 95% CI of the mean = [0.89, 2.60]; 577 [487, 681] units per bank per probe) on the day after implantation, decreased over the subsequent days, and stabilized after approximately 1 week, to roughly one half the number of units that could be initially recorded (8 to 120 days after implant: 0.73 [0.64, 0.83] per electrode; 281 [245, 319] per bank per probe; *Figure 4D–E*). After this initial loss during the first week, the number of total units, as well as the number of SUs, remained stable for up to 4 months, the longest time span for which we have sufficient data (ANOVA, single-factor (days), for only data > 7 days: tested on the number of units p=*0.895* and on the number of SUs: p=*0.581*). The fraction of units that were SUs did not change over time (ANOVA, p=*0.331*). The activity levels of the isolated units and the size of the spike

**Table 1.** All implantations.

Note that rats A230, A241, and A243 had multiple probes implanted simultaneously. A positive angle in the coronal plane indicates that the probe tip was more lateral than the insertion site at the brain surface, and a positive angle in the sagittal plane indicates that the probe tip was more anterior than the insertion site. [1] Could not be successfully explanted, most likely because no petroleum jelly was applied at the base of the implant to mitigate blood entering into the space between the holder and the chassis and bonding them together. [2] Implant detached before explantation could be attempted. This only occurred for rats that had undergone multiple sequential surgeries, and only after 100 days or more from the initial surgery. Skull degradation was observed in these cases. [3] Probe was damaged during recording before explantation could be attempted.

| Implant # | Date implanted | Animal ID | Probe serial number | Coordinates of the insertion site relative to bregma AP (mm) | ML (mm) | Insertion depth (mm) | Angle in the coronal plane (°) | Angle in the sagittal plane (°) | Shank plane angle relative to the sagittal plane (°) | Animal's age at the time of implant (day) | Number of times the probe was previously implanted | Holder version | Explantation attempted | Reusable after explantation |
|---|---|---|---|---|---|---|---|---|---|---|---|---|---|---|
| 1 | 4/6/18 | T176 | 619040938 | 4 | 1 | 4.2 | −10 | 0 | 90 | 493 | 0 | Early | Yes, 96 days post-implant | No [1] |
| 2 | 5/24/18 | T181 | 17131306102 | 1.9 | 1.3 | 8.2 | 15 | 0 | 0 | 238 | 0 | | Yes, 536 days post-implant | |
| 3 | 5/30/18 | T182 | 17131311881 | 1.9 | 1.3 | 8.2 | 15 | 0 | 0 | 244 | 0 | | Yes, 125 days post-implant | Yes |
| 4 | 1/20/19 | T179 | 17131311342 | −7 | 3 | 5.8 | 0 | 0 | 90 | 505 | 0 | | Yes, 176 days post-implant | No [3] |
| 5 | 4/22/19 | T196 | 17131312042 | −7.2 | 1.7 | 8 | −40 | 0 | 90 | 410 | 0 | | Yes, 135 days post-implant | Yes |
| 6 | 5/6/19 | T209 | 17131311352 | −7.4 | 1.6 | 7.8 | −40 | 0 | 90 | 370 | 0 | | Yes, 122 days post-implant | Yes |
| 7 | 5/20/19 | A242 | 17131311621 | −2.35 | 4.95 | 7.6 | 5 | 0 | 0 | 258 | 0 | | No [3] | — |
| 8 | 5/22/19 | K265 | 17131311562 | 1.9 | 0.8 | 7 | 20 | 0 | 90 | 656 | 0 | | Yes, 53 days post-implant | No [1] |
| 9 | 7/2/19 | A230 | 17131308411 | 2.2 | 5 | 8.6 | 5 | 0 | 0 | 420 | 0 | Current | No [3] | — |
| 10 | 7/2/19 | A230 | 17131308571 | 0.8 | 4 | 6.6 | −2 | 0 | 0 | 420 | 0 | | | |
| 11 | 7/2/19 | A230 | 18005106831 | 4 | 0.5 | 7.5 | 26 | −29 | 45 | 420 | 0 | | | |
| 12 | 8/3/19 | T212 | 17131312432 | 4 | 1 | 4.2 | −10 | 0 | 90 | 459 | 0 | | Yes, 31 days post-implant | Yes |
| 13 | 9/11/19 | A241 | 18194823302 | −0.6 | 4 | 10 | −2 | 0 | 0 | 372 | 0 | | Yes, 140 days post-implant | Yes |
| 14 | 9/11/19 | A241 | 18194823631 | 0.7 | 2.15 | 10 | 2 | 0 | 0 | 372 | 0 | | No [3] | — |
| 15 | 9/13/19 | A243 | 18194823211 | −0.6 | 4 | 9.45 | −2 | 0 | 0 | 374 | 0 | | No [3] | — |
| 16 | 9/13/19 | A243 | 18194824132 | 0.7 | 2.45 | 9 | 0 | 0 | 0 | 374 | 0 | | | |
| 17 | 11/14/19 | T224 | 17131312432 | 4 | 1 | 4.2 | −10 | 0 | 90 | 520 | 1 | | Yes, 81 days post-implant | Yes |
| 18 | 11/17/19 | T219 | 18194824092 | 1 | 2.4 | 8 | 0 | 15 | 90 | 523 | 0 | | No [2] | — |
| 19 | 11/22/19 | T223 | 19051017162 | 1 | 2.4 | 7.9 | 0 | 15 | 90 | 528 | 0 | | | |
| 20 | 2/4/20 | A249 | 18194819132 | 2.2 | 2.1 | 6.8 | 0 | −5 | 90 | 393 | 0 | | Recording ongoing | — |
| 21 | 2/6/20 | T249 | 17131312432 | 4 | 1.2 | 4.2 | −10 | 0 | 90 | 338 | 2 | | Yes, 39 days post-implant | Yes |
| 22 | 3/14/20 | T227 | 18194819542 | 1 | 2.4 | 8.4 | 0 | 15 | 90 | 557 | 0 | | No [2] | — |

**Table 2.** The brain areas recorded in each implant.

The implant numbers are the same as in **Table 1** and in **Figure 4—figure supplement 4**. No recording was obtained from implants #9, 11, 15 due to poor signal quality.

| Brain area | Implant # | Animal ID | Probe serial number | Animal's age on the time of implant (day) | Number of times the probe was previously implanted | Insertion depth (mm) | Number of electrodes in the brain area | Shank plane angle relative to the sagittal plane (°) | Center of mass of electrodes in the brain area (mm) | | |
|---|---|---|---|---|---|---|---|---|---|---|---|
| | | | | | | | | | AP | ML | DV |
| Dorsomedial frontal cortex | 1 | T176 | 619040938 | 493 | 0 | 4.2 | 170 | 0 | 4 | 0.8 | -1.1 |
| | 12 | T212 | 17131312432 | 459 | 0 | 4.2 | 174 | 0 | 4 | 0.8 | -1 |
| | 17 | T224 | 17131312432 | 520 | 1 | 4.2 | 174 | 0 | 4 | 0.8 | -1 |
| | 21 | T249 | 17131312432 | 338 | 2 | 4.2 | 186 | 0 | 4 | 1 | -1.1 |
| Ventromedial frontal cortex | 1 | T176 | 619040938 | 493 | 0 | 4.2 | 213 | 0 | 4 | 0.5 | -3 |
| | 12 | T212 | 17131312432 | 459 | 0 | 4.2 | 209 | 0 | 4 | 0.5 | -2.9 |
| | 17 | T224 | 17131312432 | 520 | 1 | 4.2 | 209 | 0 | 4 | 0.5 | -2.9 |
| | 21 | T249 | 17131312432 | 338 | 2 | 4.2 | 197 | 0 | 4 | 0.7 | -3 |
| Motor cortex | 2 | T181 | 17131306102 | 238 | 0 | 8.2 | 214 | 90 | 1.9 | 1.7 | -1.4 |
| | 3 | T182 | 17131311881 | 244 | 0 | 8.2 | 214 | 90 | 1.9 | 1.7 | -1.4 |
| | 14 | A241 | 18194823631 | 372 | 0 | 10 | 27 | 90 | 0.7 | 2.2 | -2.2 |
| | 16 | A243 | 18194824132 | 374 | 0 | 9 | 48 | 90 | 0.7 | 2.4 | -1.4 |
| | 18 | T219 | 18194824092 | 523 | 0 | 8 | 241 | 0 | 1.3 | 2.4 | -1.3 |
| | 19 | T223 | 19051017162 | 528 | 0 | 7.9 | 241 | 0 | 1.3 | 2.4 | -1.2 |
| | 22 | T227 | 18194819542 | 557 | 0 | 8.4 | 241 | 0 | 1.4 | 2.4 | -1.7 |
| Dorsomedial striatum | 2 | T181 | 17131306102 | 238 | 0 | 8.2 | 198 | 90 | 1.9 | 2.3 | -3.8 |
| | 3 | T182 | 17131311881 | 244 | 0 | 8.2 | 236 | 90 | 1.9 | 2.3 | -3.7 |
| | 8 | K265 | 17131311562 | 656 | 0 | 7 | 234 | 0 | 1.9 | 2.2 | -3.9 |
| | 10 | A230 | 17131308571 | 420 | 0 | 6.6 | 184 | 90 | 0.8 | 3.9 | -3.5 |
| | 13 | A241 | 18194823302 | 372 | 0 | 10 | 199 | 90 | -0.6 | 3.9 | -3.7 |
| | 14 | A241 | 18194823631 | 372 | 0 | 10 | 219 | 90 | 0.7 | 2.3 | -3.5 |
| | 16 | A243 | 18194824132 | 374 | 0 | 9 | 250 | 90 | 0.7 | 2.4 | -3.3 |
| | 18 | T219 | 18194824092 | 523 | 0 | 8 | 270 | 0 | 2 | 2.4 | -3.8 |
| | 19 | T223 | 19051017162 | 528 | 0 | 7.9 | 270 | 0 | 2 | 2.4 | -3.7 |
| | 20 | A249 | 18194819132 | 393 | 0 | 6.8 | 256 | 0 | 1.8 | 2.1 | -4 |
| | 22 | T227 | 18194819542 | 557 | 0 | 8.4 | 270 | 0 | 2.1 | 2.4 | -4.2 |
| Nucleus accumbens | 2 | T181 | 17131306102 | 238 | 0 | 8.2 | 191 | 90 | 1.9 | 2.9 | -5.9 |
| | 3 | T182 | 17131311881 | 244 | 0 | 8.2 | 176 | 90 | 1.9 | 2.9 | -5.9 |
| | 8 | K265 | 17131311562 | 656 | 0 | 7 | 149 | 0 | 1.9 | 2.9 | -5.7 |
| | 18 | T219 | 18194824092 | 523 | 0 | 8 | 255 | 0 | 2.7 | 2.4 | -6.3 |
| | 19 | T223 | 19051017162 | 528 | 0 | 7.9 | 255 | 0 | 2.7 | 2.4 | -6.2 |
| | 20 | A249 | 18194819132 | 393 | 0 | 6.8 | 127 | 0 | 1.7 | 2.1 | -5.9 |
| | 22 | T227 | 18194819542 | 557 | 0 | 8.4 | 255 | 0 | 2.8 | 2.4 | -6.7 |
| Piriform areas | 2 | T181 | 17131306102 | 238 | 0 | 8.2 | 115 | 90 | 1.9 | 3.2 | -7.2 |
| | 3 | T182 | 17131311881 | 244 | 0 | 8.2 | 106 | 90 | 1.9 | 3.2 | -7.2 |
| Primary somatosensory cortex | 7 | A242 | 17131311621 | 258 | 0 | 7.6 | 353 | 90 | -2.3 | 5.1 | -1.8 |
| | 13 | A241 | 18194823302 | 372 | 0 | 10 | 58 | 90 | -0.6 | 3.9 | -2.4 |
| Ventral pallidum | 16 | A243 | 18194824132 | 374 | 0 | 9 | 99 | 90 | 0.7 | 2.5 | -6.6 |

*Table 2 continued on next page*

*Table 2 continued*

| Brain area | Implant # | Animal ID | Probe serial number | Animal's age on the time of implant (day) | Number of times the probe was previously implanted | Insertion depth (mm) | Number of electrodes in the brain area | Shank plane angle relative to the sagittal plane (°) | Center of mass of electrodes in the brain area (mm) | | |
|---|---|---|---|---|---|---|---|---|---|---|---|
| | | | | | | | | | AP | ML | DV |
| Amygdaloid complex | 7 | A242 | 17131311621 | 258 | 0 | 7.6 | 70 | 90 | -2.4 | 5.6 | -7 |
| | 13 | A241 | 18194823302 | 372 | 0 | 10 | 208 | 90 | -0.6 | 3.7 | -8.7 |
| | 16 | A243 | 18194824132 | 374 | 0 | 9 | 69 | 90 | 0.7 | 2.5 | -8.5 |
| Striatum tail | 7 | A242 | 17131311621 | 258 | 0 | 7.6 | 209 | 90 | -2.3 | 5.4 | -5.1 |
| Dorsal-posterior cortex | 4 | T179 | 17131311342 | 505 | 0 | 5.8 | 109 | 0 | -7 | 3 | -0.9 |
| Postsubiculum | 4 | T179 | 17131311342 | 505 | 0 | 5.8 | 112 | 0 | -7 | 3 | -2.4 |
| Superior colliculus | 5 | T196 | 17131312042 | 410 | 0 | 8 | 239 | 0 | -7.2 | 1.6 | -4 |
| | 6 | T209 | 17131311352 | 370 | 0 | 7.8 | 190 | 0 | -7.4 | 1.4 | -3.6 |
| Posterior tectum | 4 | T179 | 17131311342 | 505 | 0 | 5.8 | 259 | 0 | -7 | 3 | -4.3 |
| | 5 | T196 | 17131312042 | 410 | 0 | 8 | 144 | 0 | -7.2 | 2.9 | -5.4 |
| | 6 | T209 | 17131311352 | 370 | 0 | 7.8 | 193 | 0 | -7.4 | 2.7 | -5.1 |
| Globus pallidus | 16 | A243 | 18194824132 | 374 | 0 | 9 | 139 | 90 | 0.7 | 2.5 | -5.4 |
| Bed nucleus of the stria terminalis | 14 | A241 | 18194823631 | 372 | 0 | 10 | 221 | 90 | 0.7 | 2.3 | -5.7 |
| Preoptic area | 14 | A241 | 18194823631 | 372 | 0 | 10 | 299 | 90 | 0.7 | 2.4 | -8.3 |
| | 16 | A243 | 18194824132 | 374 | 0 | 9 | 100 | 90 | 0.7 | 2.5 | -7.6 |

waveforms also remained stable (*Figure 4—figure supplement 5*; ANOVA, event rate: p=*0.188*; peak-to-peak amplitude: p=*0.290*). These yields indicate that our approach provides sufficient signal for studying coordinated, population-level activity during behavior. Furthermore, unit yields could, in principle, be increased beyond what we report here by selecting recording sites in a way that takes advantage of the observed distribution of units across the probe (*Choi et al., 2020*).

The loss of units over time can be described quantitatively as the sum of two exponentially decaying terms (Materials and methods), which can be interpreted as two subpopulations with different time constants of disappearance. Model estimates (*Figure 4—figure supplement 6*) indicate that half of the population (units: 0.50 bootstrapped 95% CI = [0.38, 0.63]; SUs only: 0.48 [0.26, 0.61]) was associated with a fast exponential change rate (units: $-0.45$ [$-10^6$,$-0.13$]; SUs: $-1.51$ [$-10^6$,$-0.31$]). A change rate of $\sim-0.5$ (equivalent to a half-life of $\sim$1.4 days) indicates that within a week of the implant, this subpopulation declined to less than 5% of its initial value, thereby accounting for the rapid decrease in yield within the first week. The subpopulation with faster decay might be neurons that were acutely damaged during probe insertion (*Bjornsson et al., 2006*). The remaining subpopulation had a change rate that was nearly zero: $-0.0001$ [$-0.0025$, $-10^{-6}$]; SUs: $-0.0005$ [$-0.0025$,$-0.10^{-6}$], accounting for the stable yield after the first week.

We further sought to identify the experimental factors, including the anatomical positions and the orientation of the probe, that might affect the time course of unit loss. Signal stability depended on anatomical position, especially on the dorsoventral (DV) position in the brain (*Figure 4G–I*). The number of units recorded from the electrodes in the most superficial two millimeters of brain tissue (corresponding to the dorsal cerebral cortex in all our recordings) declined steadily over several months until a near-complete disappearance of units. The ratio of the average number of units > 30 days after implantation to the average number of units on the first day after implantation was 0.18 ([0.10, 0.29]; SUs only: 0.17 [0.07, 0.30]). In contrast, the electrodes more than two millimeters below the brain surface, which targeted the striatum and ventral medial frontal cortex in the majority of implants, were associated with a significantly less degradation that eventually stabilized. The ratio of

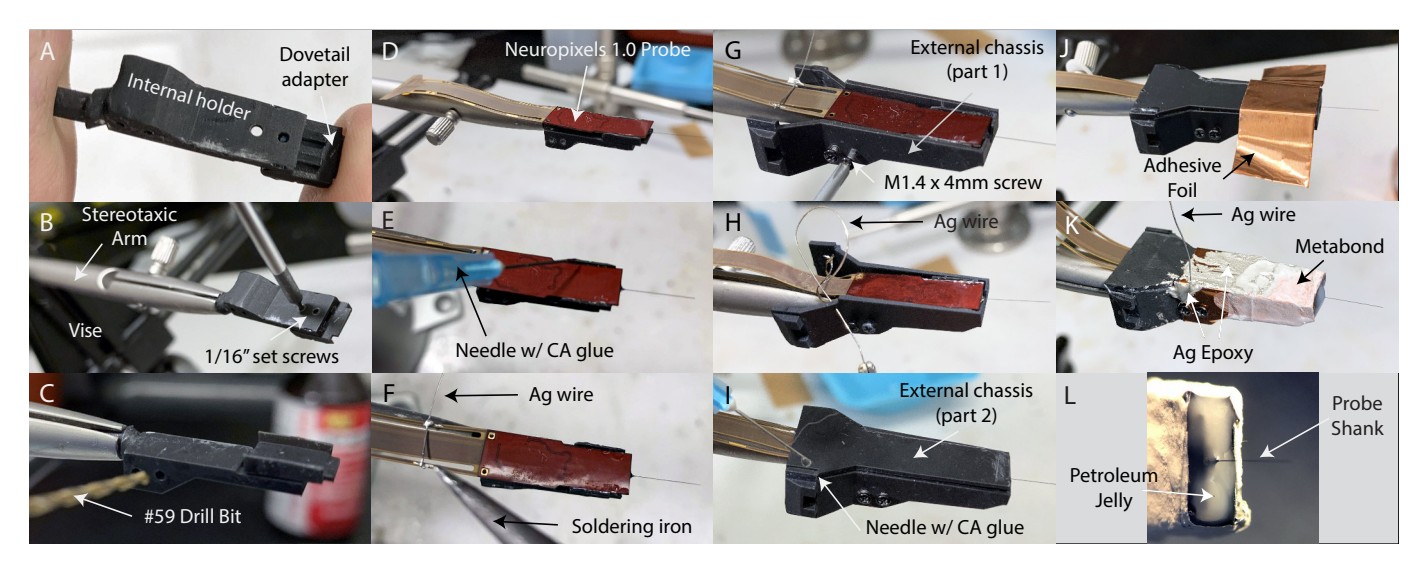

**Figure 2.** Implant construction. (A) The internal holder and dovetail adapter are slid onto one another with a dovetail joint. (B) The internal holder is held in place during subsequent construction using a stereotaxic arm mounted on a tabletop with a vise. The internal holder and dovetail adapter are secured together with set screws. (C) The screw holes used to attach the internal holder to the external chassis are drilled out by hand with a pin vise, and the entire assembly is rotated such that the dovetail adapter's flat platform is facing up. (D) The probe is laid flat against the platform, with the silicon spacer facing down. (E) The probe is glued in place by dropping several small drops of thin-viscosity cyanoacrylate (CA) glue in the gaps between the probe and platform. (F) A 10 cm length of bare silver (Ag) wire is soldered to pads on the flex cable to electrically connect the probe's ground and external reference. (G) The first part of the external chassis is screwed onto the internal holder. (H) The silver wire is threaded through a small hole in the external chassis so that it can be connected to the animal ground during surgery. Note that this panel is mirror-reversed relative to the other panels to better illustrate the silver wire. (I) The second part of the external chassis is laid on top and glued to the first part with several drops of CA glue. (J) Adhesive copper foil tape is wrapped around the ventral portion of the assembly to shield the probe and seal any gaps between the two parts of the external chassis. (K) The implant assembly, fully constructed. The silver wire is electrically connected to the copper shield with silver epoxy. Silver epoxy is also applied where the copper shield overlapped itself, to ensure the adhesive backing did not prevent electrical contact. C and B Metabond (Parkell) is applied to the portion of the assembly to which dental cement will be applied during surgery (*Figure 3B*), to improve adhesion. Note that, like panel H, this panel is mirror-reversed for a better view of the silver wire. (L) The face of the implant from which the probe shank protrudes is sealed with petroleum jelly. The petroleum jelly is applied in liquid drops from a cautery.

the average number of units > 30 days after implantation to the average number of units on the first day was 0.50 [0.38, 0.63] (SUs only: 0.54 [0.40, 0.71]). Fitting the sum-of-exponentials model separately to data from either the superficial and deep electrodes, there was a significant difference in the change rates of the more slowly decay subpopulation (p=0.002, bootstrap test; SUs: p=*0.001*; *Figure 4G–I*; and *Figure 4—figure supplement 6*). This led to a significant difference in the model's effective time constant, which is the number of days until 1/e, or 37%, of the units remained, between the superficial and deep electrodes (*Figure 4I*; superficial units: 19 [5, 44], deep units: $10^6$ [377, $10^6$]; superficial SUs only: 13 [1, 59], deep SUs only: 1477 [280, $10^6$]). These results indicate that long-term stability depends on dorsoventral location.

We also observed a difference in stability between our recordings anterior and posterior to Bregma (*Figure 4J–L*). Recordings from more posterior sites were associated with fewer initial units (p=*0.010*, rank-sum test), which could be in part due to a lower neuron density in the posterior structures compared to the anterior structures that we recorded from *Erö et al., 2018*. There was also a more severe decay in yield in the posterior areas, with a virtually complete loss of units after one month. The ratio of the total number of units 30 days or more after implantation to the unit count 1 day after implantation was 0.09 [0.04, 0.22] for posterior recordings and 0.50 [0.40, 0.63] for the anterior recordings. The same ratio for the number of SUs only was 0.06 [0.00, 0.21] for posterior recordings and 0.53 [0.38, 0.72] for anterior recordings. Fitting the sum-of-exponentials model to this data, there was a significant difference in the change rate of the more slowly decay

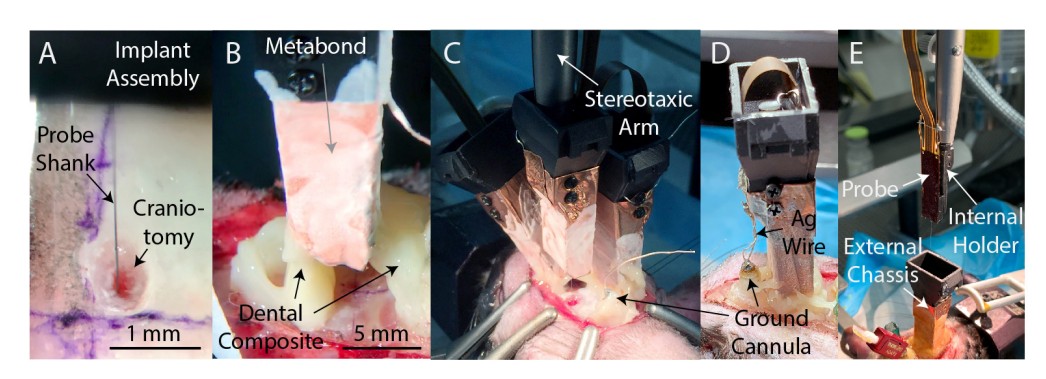

**Figure 3.** Implantation and explantation. (**A**) A photograph of a probe being inserted into a craniotomy. (**B**) A photograph of a seal being formed around an implanted probe with dental composite (Absolute Dentin, Parkell). (**C**) A photograph of a third probe being lowered into the brain of subject A230. Note the steel ground cannula at the posterior edge of the exposed skull, soldered to a length (~5 cm) of silver wire and fixed in place with dental composite. (**D**) After implantation of all probes, the silver wire soldered to the ground cannula is soldered to the silver wires soldered to each probe, to provide a common ground for all probes. Dental acrylic (Duralay, Reliance Dental) was then applied to fully encapsulate the implant, including the silver wires. (**E**) A photograph of the explantation of a probe. The probe and internal holder have been lifted free from the implant using a stereotaxic arm while the external chassis remains bonded to the skull. Also visible is an interconnect for an optical fiber (red).

subpopulation (p=0.002, bootstrap test; SUs: p=*0.001*) and also in the initial unit count(p=*0.005*; SUs: p=*0.003*; *Figure 4J–L*; and *Figure 4—figure supplement 6*). The effective time constant of the sum-of-exponentials model was significantly different between units recorded from anterior and posterior electrodes (*Figure 4L*; anterior units: 527 [122, $10^6$]; posterior units: 9 [3, 22]; anterior SUs only: 359 [92, $10^6$], posterior SUs only: 10 [1, 32]). These results indicate that long-term stability also depends on anteroposterior (AP) location.

In addition to DV and AP positions, there are likely other factors that also affect the time course of unit loss, such as the ML position. In addition to anatomical position, the position of a recording site on the probe shank might be relevant because the probe is more flexible near its tip. Greater flexibility allows the shank to move more easily with the surrounding brain tissue and therefore may lead to less tissue damage over time (*Lecomte et al., 2018*). The shank orientation (i.e. the angle of the probe's plane relative to the brain) might also be relevant because the probe's width is three times its thickness (70 μm vs. 24 μm), making it ~ 27 times more flexible in the axis perpendicular to its plane. If the relative motion between the probe and the brain is anisotropic, the orientation of the probe's more flexible axis could affect yield stability.

To infer the relationship between these experimental factors and the unit count over time, the sum-of-exponentials model was next fitted using all these factors as regressors (*Figure 5*; referred to as the sum-of-exponentials regression (SoER) model to distinguish it from the simpler variant that does not include any regressor). To take advantage of the larger sample size and the greater variation in the experimental factors across individual electrodes, the SoER model was fitted to the unit count of each electrode in each recording session (Materials and methods). The model assumes two subpopulations of units (α and *1-α*) whose relative sizes are constant across all recordings. The change rates of the two subpopulations at each electrode are described as an identical linear combination of the regressors, up to an offset unique to each population ($\beta_{fast}$ for the α population and $\beta_{slow}$ for the *1-α* population). In this way, the change rates of the two subpopulations could vary across electrodes while exhibiting a constant difference ($\beta_{fast}$ - $\beta_{slow}$). The regressors included AP (mm anterior), DV (mm relative to brain surface), ML (mm from midline), SP (shank position, mm above tip), and SO (the orientation of the shank's plane relative to the brain's sagittal plane and has the range [0°, 90°]). The initial unit count ($N_1$) was also a linear combination of the regressors, excluding SP and SO because their contribution to the initial yield is not readily interpretable. The three parameters (α, $\beta_{fast}$, and $\beta_{slow}$) were fixed across electrodes to limit the number of parameters.

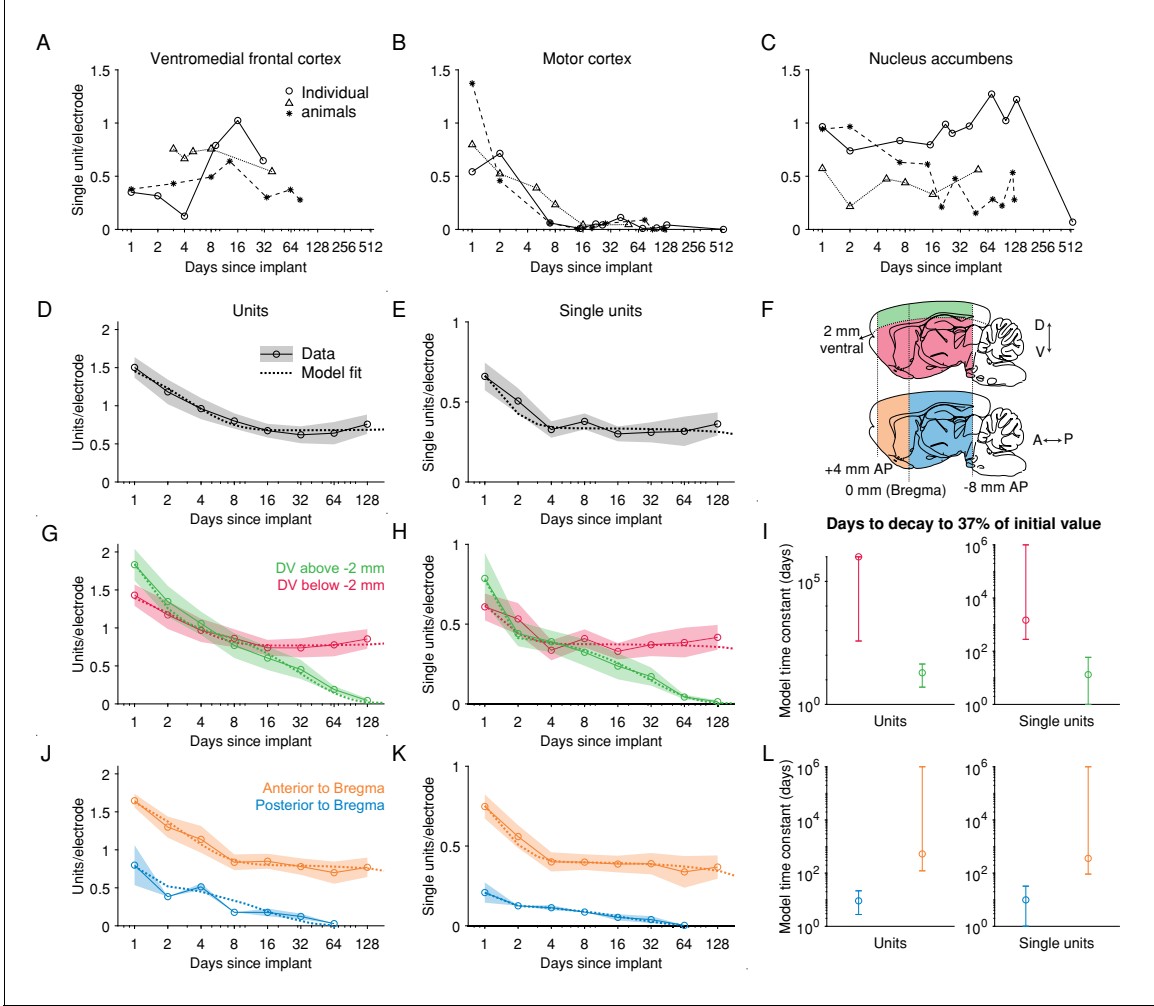

**Figure 4.** After an initial loss of units, spiking signals can be maintained >60 days in anterior, deeper brain regions. (**A**) Recordings from the medial prefrontal cortex (encompassing the areas labelled in the Paxinos Brain Atlas as prelimbic cortex and medial orbital cortex) across three example animals. Each combination of marker and line types indicates one animal. (**B**) Recordings from motor cortex. (**C**) Recordings from nucleus accumbens. (**D**) The number of units recorded per electrode per session. Shading represents mean +/- 1 s.e.m. across recording sessions. The dashed line is the fit of a sum of two exponential decay terms, representing two subpopulations with different time constants of decay. (**E**) The number of single units. (**F**) To explore the dependence of signal stability on anatomical position, units were separated into two groups along either the dorsoventral (DV) axis or the anteroposterior (AP) axis. (**G**) The number of units recorded either more superficial to or deeper than 2 mm below the brain surface, normalized by the number of electrodes in the same region. (**H**) Similar to G, but showing the number of single units. (**I**) The model time constant is the inferred number of days after implantation when the count of units (or single units) declined to $1/e$, or ~37%, of the count on the first day after implant. The 95% confidence intervals were computed by drawing 1000 bootstrap samples from the data. (**J-L**) Similar to G-I, but for data grouped according to their position along the anterior-posterior axis of the brain. (D-E) N = [12, 8, 20, 18, 32, 20, 13, 16] recording sessions for each bin. (G-H) DV [−10,−2] mm: N = [12, 8, 19, 18, 32, 20, 13, 16]; DV [−2, 0] mm: N = [6, 4, 11, 7, 14, 10, 8, 5]. (J-K) AP [−8, 0] mm: N = [2, 1, 5, 1, 8, 4, 1]; AP [0, 4] mm: N = [10, 7, 15, 17, 24, 16, 12, 16].

The online version of this article includes the following figure supplement(s) for figure 4:

**Figure supplement 1.** Example histological images of probe tracks.

**Figure supplement 2.** Comparison of yields before and after manual curation.

**Figure supplement 3.** No degradation of spiking signals was detected over two months in rat medial prefrontal cortex (mPFC), the brain region in which the stability of spiking signals was examined in *Jun et al., 2017*.

**Figure supplement 4.** The yield in single units over time for each brain area and each implant.

**Figure supplement 5.** The event rate and peak-to-peak amplitudes.

**Figure supplement 6.** Coefficient estimates of the sum-of-exponentials model used to describe unit loss over time.

Because the regressors are partly correlated, estimation depends on what other regressors are included in the model. To make the inference more reliable, all-subset variable selection was performed to exhaustively identify the subset of regressors that best predicted the data (*Figure 5A*). We also included model variants in which we substituted the continuous regressor *DV* with *DV>-2mm*, which approximately separates dorsal cerebral cortex from deeper areas of the brain in our recordings, to examine whether unit count is better predicted by whether an electrode is in dorsal cortex rather than by its depth.

Among 448 variable subsets (Materials and methods), the five model variants with the lowest out-of-sample log-likelihood (LL) included the *AP* and *ML* variables as predictors for the initial unit count ($N_1$) (*Figure 5B*), such that a larger initial unit count was associated with a more anterior and a more medial position. The best model also included the regressors *AP*, *DV>-2*, and *SO* (shank orientation) for the change rate term ($k$) (*Figure 4O*), such that a slower loss of units was associated with an electrode that was more anterior, below dorsal cortex, and on a shank whose plane is parallel to the brain's sagittal plane. The coefficients in the change rate term were similar in the order of magnitude to the that of baseline change rate of the more slowly decaying subpopulation ($\beta_{slow}$), indicating that the experimental factors tested here affected the slow disappearance of units over multiple weeks and months rather than the rapid disappearance during the first few days after implantation. If any coefficient had a magnitude that is intermediate between $\beta_{fast}$ and $\beta_{slow}$, then it would suggest that the experimental factors affected the fast and slowly decaying subpopulations differently, rather than in the same way, as it is assumed by the model. Finally, the model variant without any of the experimental factors as regressors (i.e. with only the constant terms) was ranked 396 out of 448, indicating that the factors are relevant for predicting unit loss over time.

The best model included five (non-constant) regressors. Examining each group of model variants with four, three, two, and one regressors revealed that the regressors in the best *n*-regressor model is a superset of the regressors in the best (*n-1*)-regressor model. This nesting of regressors is not required by the all-subset variable selection process (such as when the regressors are shuffled), and it provides a list of regressors ordered by decreasing importance for predicting unit count across days: $N_1$-*AP*, *k-DV>-2*, $N_1$-*ML*, *k-AP*, *k-SO*. Taken together, the results indicate that a higher initial unit count is associated with electrodes that are more anterior and medial, and that slower decay is associated with electrodes that are below dorsal cortex, further anterior, and on a shank parallel to the sagittal plane.

The results of the SoER model may depend on the assumption that the relative proportion of fast-decaying and the slow-decaying populations is fixed or on the particular set of regressors tested. To confirm the results from the modeling, we fitted an elaborated SoER model (*Figure 5—figure supplement 1*; Materials and methods) whose fast-decaying and the slow-decaying populations are entirely independent and depend on separate linear combinations of the regressors. The elaborated model has two additional regressors, the number of the times a probe has been previously implanted (0, 1, or 2) and the animal's age (in days). Because age can affect neuronal biophysical properties and action potential shape (*Tripathy et al., 2015*) and neuroimmune reactivity (*Clarke et al., 2018*), it may affect the yield of chronic recording. Due to the larger number of coefficients in the elaborated SoER model (21 as opposed 14 coefficients in the basic SoER model), parameter selection was performed using a L1 regularization. When both the elaborated and basic SoER models were fitted using L1 regularization, the elaborated model has a modestly higher out-of-sample log-likelihood than that of the basic model (log-likelihood ratio = 1.006) and has a lower Bayesian Information Criterion ($\Delta$BIC = $-202$). The coefficient estimates of the elaborated SoER model indicate that the relative proportion of fast-decaying and slow-decaying units depends on *AP*, *ML*, and age. The coefficients also indicate older animals are associated with a lower initial unit count but a slower rate of decay, and previous uses of the same probe are associated with slightly faster decay (though the latter results depend on only two implants with used probes). Importantly, the coefficient estimates of the elaborated model corroborate the results of the basic model by demonstrating a dependence of the initial unit count on the *AP* and *ML* regressors and the change rates on the *AP, DV>-2*, and *SO* regressors.

Finally, the loss of spiking signals that we observe at first appears to contrast with a previous study reporting chronic recording in rat medial prefrontal cortex (mPFC) (*Jun et al., 2017*), where no such loss was found. However, when we examine just the mPFC recordings in our dataset (prelimbic cortex and medial orbital cortex, n = 4 rats), no degradation in the number of total units was

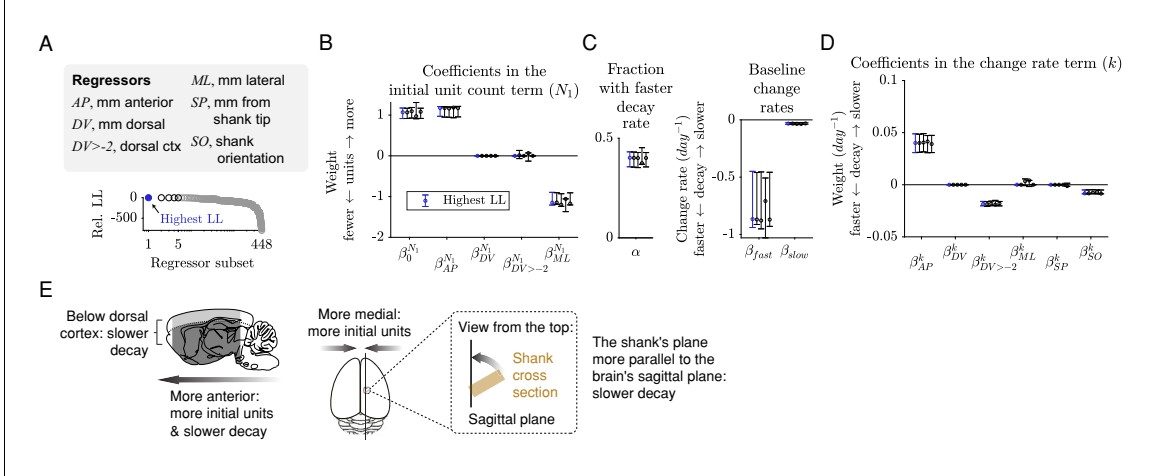

**Figure 5.** The initial unit count depends on the AP and ML positions, and the rate of decay depends on the AP position, whether an electrode is in dorsal cortex, and the shank orientation. (**A**) A sum-of-exponentials regression (SoER) model was fit to the number of units recorded from each electrode in each session (N = 57,586 recordings) to infer the relationship between experimental factors and unit loss over time. Continuous regressors *AP*, *DV*, *ML*, *SP,* and *SO* indicate an electrode's position in millimeters anterior, dorsal, lateral, and from the shank tip, and the orientation (in degrees) between the shank's plane from the brain's sagittal plane, respectively. Categorical regressors *DV>-2* indicate whether an electrode is in the dorsal cortex. Model variants with different subsets of regressors were ordered by relative out-of-sample log-likelihood (LL). The five subsets with the highest LL are shown in subsequent panels. (**B**) Coefficients in the equation term indicating the initial unit count ($N_1$) from the five regressors subsets with the highest LL. Initial unit count consistently depends on *AP* and *ML* (orange), which are included and significantly nonzero in the top five models. Error bars indicate 95% bootstrap confidence intervals. The range of all regressors is normalized to be [0,1] to facilitate comparison. The original range of the regressors were *AP* [−7.40, 4.00], *DV* [−9.78,−0.01], *ML* [0.29, 5.59], *SP* [0.02, 7.68], *SO* [0, 90]. (**C**) About 40% ($\alpha$) of the units disappeared rapidly with a baseline change rate of −0.87 ($k_{fast}$), and the remaining disappeared more slowly with a baseline change rate of −0.03 ($k_{slow}$). (**D**) Change rates depended consistently on the regressors *AP*, *DV>-2* (whether the unit was in dorsal cortex), and *SO* (angle between the shank's plane and the brain's sagittal plane), which are included and have a significantly nonzero coefficient in the top five models. (**E**) A graphical summary of the modeling results. The online version of this article includes the following figure supplement(s) for figure 5:

**Figure supplement 1.** The results of the sum-of-exponentials regression (SoER) model are corroborated by the results of an elaborated SoER model.

apparent over a 2-month span, and the number of SUs and event rate actually increased (*Figure 4— figure supplement 3*). This finding further demonstrates the dependence of signal stability on anatomical AP and DV positions, because our recordings from mPFC were the most anterior (AP = 4.0 mm from Bregma) and were taken 1.8 mm or more below the brain surface.

## Behavioral performance while tethered for recording

Our motivation to chronically implant Neuropixels probes was to study coordinated neural activity during perceptual decision-making. Therefore, we sought to determine in what ways the recording procedure might change the animals' behavioral performance. In our approach, the cable connecting the headstage and the IMEC base station was encased in a split corrugated sleeving to limit the number of turns an animal could make in a single direction to three to five and thereby avoid small bending radiuses in the cable during hours of unsupervised recording (*Figure 1D–F*). We sought to quantify the effects of the tether's movement restriction on performance.

During the perceptual decision-making task, rats maintained their nose in a center port while two streams of auditory clicks played from loudspeakers on their left and right (*Brunton et al., 2013*). When the auditory stimulus ended, rats were rewarded for reporting which stream had played the greater number of clicks by placing their nose in a left or right port (*Figure 6A*). We found little or no significant degradation in a number of critical metrics of behavioral performance between when the rats were untethered and when they were tethered for recording. Across the cohort of rats, the median number of trials for which the rats were able to successfully maintain fixation in a given session remained unchanged (*Figure 6B*), indicating that the additional weight and constraint of the implant and tether did not affect the number of completed trials. The average percent correct was

actually slightly higher during tethered performance (*Figure 6C*). The performance of individual rats was further quantified using a logistic function fit to their choice and the sensory evidence in each trial (*Figure 6D*; Materials and methods). The parameter of the logistic function that quantifies sensitivity to the sensory evidence, and the bias parameter were not significantly different (*Figure 6E–F*). Interestingly, tethered performance was associated with a lower lapse rate (*Figure 6G*). The lower median lapse rate and higher percentage correct could potentially be because after the surgery, the rats performed the task more frequently while tethered than untethered and therefore are more engaged while tethered and showed higher performance. These results indicate that the tethering process did not substantially impair the rats' motivation or their ability to successfully complete a cognitively demanding task.

## Probe reuse

Previous work reported that reimplantation of two test-phase Neuropixels probes resulted in a lower event rate and signal-to-noise of detected spikes (*Juavinett et al., 2019*). However, event rate and signal-to-noise in the brain depend not only on probe performance but also on variability in the recording site itself. Therefore, it remains unclear to what extent probe performance degrades after prolonged implantation and explantation. Addressing this uncertainty is necessary for determining whether probe reusability is feasible.

To address this, we measured the change in the input-referred noise (i.e. noise divided by gain) in saline and compared these measurements between unimplanted and explanted probes (Materials and methods). We first noticed that, in explanted probes, recording sites inserted into the brain showed noise amplitude (root mean squared; RMS) roughly comparable to unimplanted probes. However, the part of the shank above the brain surface tended to contain localized regions with a high fraction of noisy (>20$\mu V_{RMS}$) recording sites (*Figure 7A,B*). This is likely due to the application of silicone elastomer and/or petroleum jelly to those parts of the shank during surgery, which insulated the recording sites and could not be fully removed by the enzymatic detergent used to clean the probes after explantation. These measurements may not accurately reflect performance of these recording sites upon re-implantation since insertion into the brain could remove the insulating material. Therefore, we confined further analysis to those recording sites that were lowered below the brain surface.

We compared the noise measured on the set of 2880 recording sites from three new probes to the set of 3664 recording sites from five explanted probes that had been lowered below the brain surface after their most recent explantation. We found slightly larger (p<$10^{-4}$, bootstrap test) median RMS noise values in explanted probes (median and 95% CI: 8.30 μV and [8.23, 8.37 μV]) compared to new probes (median and 95% CI: 8.10 μV and [8.04, 8.17 μV]) (*Figure 7C,D*). Crucially, this small increase (0.2 μV) is roughly three orders of magnitude less than the median peak-to-peak amplitude of detected spikes (188 μV; *Figure 4—figure supplement 5*). We further examined the fraction of noisy recording sites (*Figure 7E,F*) and found a small (<1%) increase in the explanted probes (0.40%; 95% CI: [0.22%, 0.73%]) versus explanted probes (1.35%; [1.03%, 1.74%]) that was also significant (p<$10^{-4}$, bootstrap test).

The small magnitude of increased noise after explantation suggested probe reuse would have a negligible impact on the quality of the neural signals. Consistent with this expectation, we found comparable neural signals after three implantations of the same probe in medial frontal cortex, with no trend of increasing degradation (*Figure 7G,H*; ANOVA, one-way (the number of previous implants), tested on the number of SUs (p=*0.362*) and the peak-to-peak amplitude (p=*0.767*)). Analysis of neuronal encoding during the decision-making task (*Figure 6A*) also indicates that qualitatively similar signals were observed across implantations. Specifically, the choice selectivity averaged across units was similar between the first and third implants (*Figure 7I*; ANOVA, one-way (the number of previous implants), tested on the peak choice selectivity across neurons, p=*0.382*). These results indicate the minor increases in noise level across repeated cycles of implantation and explantation of the same probe did not prevent acquisition of high-quality spiking signals.

We next sought to better understand the source of the observed noise and its slight increase in explanted probes. We reasoned that the degree of correlation in noise amplitude between probe 'banks' would be diagnostic. The Neuropixels probe contains fewer acquisition channels than physical recording sites, with each channel able to programmatically switch between addressing a site on bank 0 or a site on bank 1. As such, the recording sites across banks can be paired according to

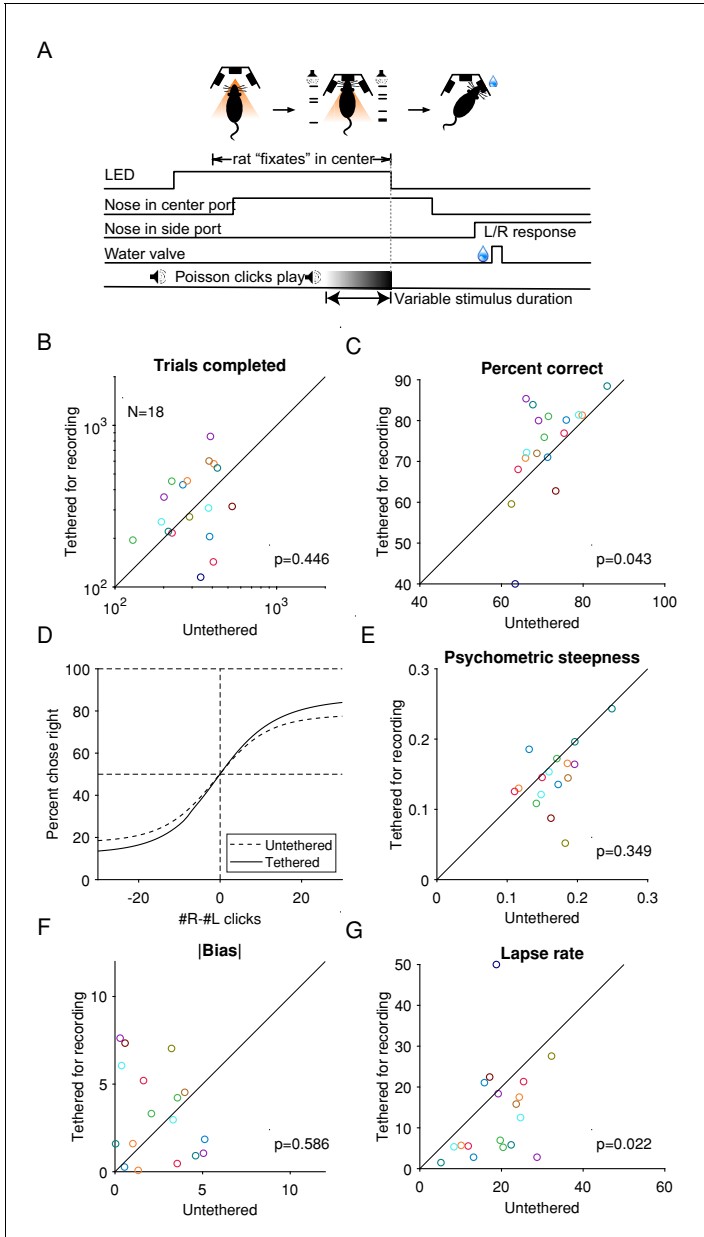

**Figure 6.** While tethered for Neuropixel recording without a cable commutator, rats performed a cognitively demanding task at a level similar to when they were untethered. (**A**) Task schematic: rats hold their nose in the center port while listening to two concurrent streams of auditory clicks, one from a loudspeaker to its left and the other to its right. At the end of the stimulus, the rat receives a reward if it oriented toward the port on the side where more clicks were played. Reproduced from *Brunton et al., 2013*. (**B**) The median number of trials completed by an animal in each training or recording session, compared between tethered and untethered. Each marker indicates an animal. p-Values are based on the null hypothesis that the median difference between paired observations is zero and calculated using the Wilcoxon signed-rank test. (**C**) The percentage of trials when an animal responded correctly. (**D**) Logistic curves fitted to data pooled across from all animals. (**E**) Comparison of the behavioral psychometric curves' sensitivity parameter, which controls the steepness of the logistic function. (**G**) Absolute value of the bias parameter. (**F**) The average of the lapse rate. A larger lapse indicates a larger fraction of trials when the animal was not guided by the stimulus or a decreased ability to take advantage of large click differences.

which acquisition channel addresses them. Noise values that are uncorrelated across these pairs presumably reflect a source intrinsic to the recording sites themselves, whereas noise values that are correlated across these pairs presumably reflect noise introduced downstream in the shared signal processing circuitry. We found that the noise across banks was highly correlated across all recording sites measured ($R^2$ = 0.89, n = 2650 recording site pairs; *Figure 7J*), but with significant variability in the degree of correlation among explanted probes (*Figure 7K*). Overall, the noise across banks was significantly higher for pairs of recording sites on new probes ($R^2$ = 0.94, 95% CI = [0.89,0.97], n = 1152 recording site pairs) than for those on explanted probes ($R^2$ = 0.84, 95% CI = [0.64,0.93], n = 1498 recording site pairs) (*Figure 7L*). Thus, the electrical noise in both new and explanted Neuropixels probes is mostly introduced downstream of the individual recording sites, but noise intrinsic to the recording sites plays a more significant role after long-term implantation. Taken together with the slightly larger RMS noise observed in explanted probes, these data support the idea that long-term implantation leads to modest, but measurable, degradation of the recording sites that both increases and decorrelates noise across the shank.

We have provided a table detailing the outcome of all probes used in these experiments, including the probes used to develop the techniques described here (*Table 1*). In early testing of three probes, we found that not applying petroleum jelly to the base of the chassis resulted in blood and cerebrospinal fluid entering the cavity of the 3D printed chassis and prevented retrieval of the holder. The implant of three other probes detached from the animal's skull before explantation could be attempted, but all three of these probes were used in a pilot experiment that implanted rats that had previously undergone the injection of virus and implantation of optical fibers at least 60 days prior. All other attempted explantations were successful. Among the explanted probes, one probe could not be reused because we changed the grounding to the recording system during data acquisition, and the probe thereafter was unresponsive. All other explanted probes were reused for recording, and one was re-implanted and explanted in two additional animals (three total implants).

## Discussion

Neuropixels probes are an exciting addition to the toolkit for chronic electrophysiology in freely moving animals, and will remain so as probe development advances and wireless acquisition systems become available. Here, we sought to address several outstanding uncertainties about the capability of Neuropixels probes for chronic use, a needed step to realize this potential. First, the expected yield over multiple months is currently known for only a single brain area (rat medial prefrontal cortex) (*Jun et al., 2017*). Second, methods currently do not exist for Neuropixels implantation that allow multiple probes to be implanted, are compatible with probe explantation, and are robust enough to ensure high yields after months on a rat or other similarly sized animals. Finally, the performance of previously explanted probes compared to unimplanted probes remains unclear. Here, inspired by previous approaches for chronic Neuropixels implants (*Juavinett et al., 2019*; *Jun et al., 2017*), we report a system for chronic implantation of Neuropixels probes in rats that is sufficiently compact for multiple probes to be simultaneously implanted, robust enough to withstand months of implantation and hours of unmonitored recording per day, and compatible with probe recovery and reuse. We validated the system in 18 rats and documented the performance of the chronic implants over many months.

Encouragingly, we found that on average, the degradation of spiking signals occurred during the first week of recording and then remained stable for up to 4 months. However, stability varied systematically according to brain region, with greatest stability in anterior and ventral brain regions. During recording, even with a restriction on movement imposed by a non-commutated tether, rats performed a cognitively demanding task at a level similar to their untethered performance before implantation. Lastly, we demonstrate that after multiple months of implantation, explanted probes show a slight decrease in performance, but this has no measurable consequence on the acquisition of spiking signals. These results provide practical considerations that can facilitate further adoption, standardization, and development of Neuropixels probes for chronic recording in unrestrained animals.

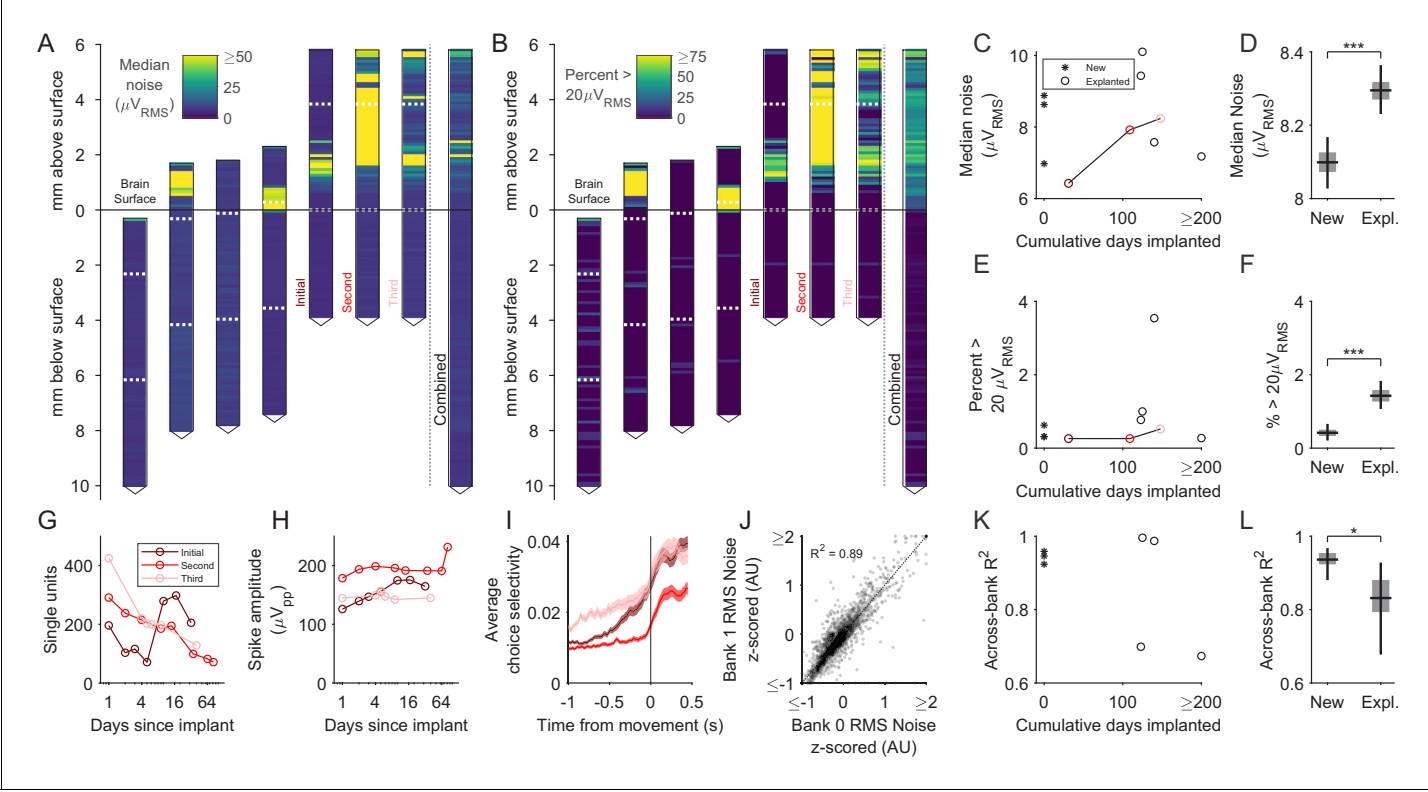

**Figure 7.** Explanted probes and unimplanted probes have similar input-referred noise and can acquire neural signals of similar quality. (A) Shank map showing the RMS input-referred noise (i.e. noise divided by gain), measured in saline, along the entire shank of explanted probes, aligned and ordered according to their implanted depth. Note high noise confined to the part of the shank above the brain surface, where silicone elastomer and/or petroleum jelly were applied during surgery. These sites are excluded from the data contributing to the subsequent panels. Combined data across explanted probes is at the right. Note that the three rightmost shank maps come from repeated explantations of the same probe. Dashed horizontal white lines indicate the boundaries between banks. (B) Same as A, but showing the percentage of recording sites with RMS noise >20μV. (C) The median noise, shown separately by probe as a function of time implanted. Measurements from one probe that was implanted in three different animals are connected by a line. (D) Box plot showing slightly higher median noise across recording sites of explanted probes compared to new probes (p<$10^{-4}$, bootstrap test for difference in median noise). Boxes and whiskers indicate the 50% and 95% bootstrap CI, respectively. Bootstrapping performed by resampling electrodes. (E) The percentage of recording sites with a noise value greater than 20μV$_{RMS}$ shown separately by probe as a function of time implanted. (F) Box plot showing that the fraction of recording sites with a noise value greater than 20μV$_{RMS}$ was slightly higher in explanted probes (p<$10^{-4}$, bootstrap test for difference in fraction of noisy electrodes). Boxes and whiskers indicate the 50% and 95% bootstrap CI, respectively. Bootstrapping performed by resampling electrodes. (G–H) Signal quality of the same probe implanted in the medial frontal cortex of three separate animals. (I) Choice selectivity averaged across units did qualitatively change across successive implants. Shading indicates mean +/- 1 s.e.m. (J) Noise was highly similar across pairs of recording sites on separate banks that were addressed by the same acquisition channel (n = 6544 recording sites across new and explanted probes). Scatter plot shows RMS noise values for electrode pairs in banks 0 and 1 that were implanted below the brain surface, z-scored in groups defined by probe and bank. Data points are shown with transparency, such that regions containing more of them appear darker. (K). The across-bank noise similarity ($R^2$), as computed in J, shown separately by probe as a function of time implanted. Note that the probe implanted three times is not included here, since bank one was never inserted into the brain. (L) Box plot showing the across-bank noise similarity ($R^2$) was lower in explanted probes (p=0.033, bootstrap test for difference in $R^2$). Boxes and whiskers indicate the 50% and 95% bootstrap CI, respectively. Bootstrapping performed by resampling electrodes.

## Holder design

Compared to a previous design for recoverable, chronic implantation of Neuropixels probes (*Juavinett et al., 2019*), we aimed for different experimental goals and therefore made different design choices. A primary goal was to ensure the probe was protected from the mechanical stresses in a multi-month implant on a large rodent. For this reason, the probe was fully enclosed with a 3D printed chassis which could be strongly cemented to the skull and dampen impacts. To maximize the number of probes that can fit on a single animal, a mount for the headstage was placed on top

of the probe rather than to its side. Additionally, the cross-section of the external chassis close to its ventral face was minimized to allow probes to be inserted at a large range of angles. This change greatly facilitated multi-probe implantations. Lastly, the chassis and the holder were held together using screws rather than acrylic. One possibility this allows is to move the probe relative to the chassis to try to dissociate the probe from glial scars, although we have not tested this extensively.

Many designs of chronic implant of silicon probes or tetrodes involve a drive to move the sensor into new tissue to obtain new signals after scar tissue has accumulated (*Bragin et al., 2000*; *Chung et al., 2017*). We opted to not use a drivable design because we found that the recording sites above the brain often show a degradation in performance (i.e. higher input-referred noise) due to coverage by the possible combination of silicone elastomer, petroleum jelly, or dried blood or cerebrospinal fluid. Unlike tetrodes or other silicon probes, which have their recording sites at the bottom of the probe, the recording sites on the Neuropixels are along the shank. Therefore, a driveable mechanism would primarily improve signals in ventral brain areas, at the expense of signals in the dorsal areas, by driving recording sites immersed in dorsal brain regions into ventral brain areas and driving recording sites above the brain, which we observed to show signal degradation, into the dorsal areas. Because our experimental goals involve simultaneous monitoring of both ventral and dorsal brain areas, we did not choose to use a driveable design. But other experiments might benefit from a driveable implant.

## Long-term stability of neural signals

The loss of isolated units over time consisted of two components, a fast decay during the first few days after implantation, and a slower change over multiple months, yielding on average over 100 single units for at least four months. The fast component of the decline in yield was highly consistent across recording sites, while the slow component was more variable. For brain regions posterior to Bregma or within dorsal cortex, yields declined steadily until a complete loss of units after several months. The rate of this steady decline also depended on shank orientation. However, we could not detect an effect of the position of an electrode on the probe shank, despite our prediction that greater flexibility of the shank toward its tip would reduce long-term tissue damage of the surrounding tissue.

Given the excellent performance of the probes upon reimplantation and dependence of yield on anatomical location, the decrease in yield must be related to changes in the neural tissue, changes which in turn must be heterogeneous across brain areas. Implantation of a chronic neural probe initiates a cascade of biochemical reactions that are part of the wound healing process. This has been divided into an acute and chronic phase (*Gulino et al., 2019*). The acute phase, lasting days or weeks, is caused by the trauma associated with the removal of the skull and meninges and insertion of a probe which ruptures the blood-brain barrier. These injuries lead to rapid neuronal cell death due to oxidative stress and activate a neuroinflammatory response mediated by glia (*Saxena et al., 2013*). This is followed by a chronic phase, lasting months or years, characterized by a further immune response and eventually the encapsulation of the probe by a glial scar.

Several features of our data suggest that neuronal cell death during the acute phase of the tissue response is likely the primary cause of the decrease in yield. First, most of the loss in yield occurred during the first week, before encapsulation is thought to occur (*Wellman and Kozai, 2017*). Second, the loss consisted primarily of a loss of isolated units, with no decrease in average spike amplitude. This suggests the loss of units both near and far from the shank, and no overall change in the conductivity of the surrounding tissue. This would be consistent with neuronal cell death but not with insulation of the probe in scar tissue.

We also observed pronounced differences in yield decline after this initial period depending on the electrode's AP position and on whether it was in dorsal cortex. To our knowledge, these findings cannot be easily explained based on the current literature. One potential mechanism is variability in the degree of ongoing injury after implantation due to motion of the brain relative to the skull (and therefore to a rigid probe anchored to the skull). The relative displacement could be particularly acute during head acceleration (*Zhou et al., 2020*), although even respiration causes some degree of micromotion, and both of these types of relative motion have been directly implicated in glial activation (*Lind et al., 2013*; *Muthuswamy et al., 2005*). Relative motion is expected to be variable across brain areas for at least two reasons. First, the amount of brain micromotion around an implant increases upon a durotomy (*Muthuswamy et al., 2005*). This may contribute to the loss of units in

superficial cortex relative to deeper structures. Second, the degree of micromotion and the response of the brain to deformation and inertial forces is heterogeneous (*MacManus et al., 2018*; *Sloots et al., 2020*; *Bayly et al., 2005*; *Budday et al., 2015*; *Jacobo et al., 2014*). The studies that have examined this have primarily sought to explain the heterogeneous effects of traumatic brain injury, so their implications for the long-term response to a chronically implanted foreign body are unclear. Nonetheless, these studies demonstrate clear heterogeneity in the mechanical properties of the brain, that may underlie differences in yield that can be expected after chronic implantation of rigid probes.

Our results suggest specific ways that long-term yield stability across the brain in chronic recordings could be improved with future Neuropixels probe designs. First, changes to the probe which reduce acute injury upon implantation, such as a smaller cross-sectional area and improved tip design, may reduce the dramatic decline in yield observed over the first week of recording. Second, increased mechanical flexibility, which reduces motion of the probe relative to the brain, may be essential for reducing the slow decline in yield we observed (*Jeong et al., 2015*; *Kim et al., 2004*). This slow decline led to a complete loss of units in dorsal cortex and posterior brain regions, limiting the utility of Neuropixels probes for experiments requiring long-term recordings in those regions. Mechanical flexibility could be improved by reducing the cross-sectional area of the probe or with changes in the shank material. However, these design goals may trade off with the need for a shank that is stiff enough to avoid buckling when being inserted into the brain.

Highly compliant neural probe designs have been recently developed which allow extremely stable long-term recordings (*Bondar et al., 2009*; *Chung et al., 2019*; *Jeong et al., 2015*; *Luan et al., 2017*), including tracking of individual units over extended periods of time. These take advantage of materials, such as microwires or conducting polymers, which are inherently less rigid than silicon shanks. Currently, these probe designs lag behind silicon probe technology in terms of channel count and are not readily reusable. Therefore, which of these approaches is best will likely depend on the particular application, until an approach exists that combines their advantages.

## Model interpretation and limitations

A plausible model of the loss of neurons over time is necessary for statistical inference on experimental factors that affect chronic stability of neural implants, but this has not been done before. We found that a single-term exponential regression model performed far worse at fitting the data compared to a model with two exponential terms. The success of the sum-of-exponentials model would be consistent with two distinct subpopulations of neurons having different rates of decay. One subpopulation disappears at a fast rate that is potentially due to acute damage during the probe insertion (*Bjornsson et al., 2006*). The other subpopulation declines at a slower rate, consistent with the chronic tissue response and ongoing damage from tissue motion or acceleration relative to the probe. The model fits indicated that the experimental factors had little effect on fast-decaying subpopulation.

However, we have no direct evidence that there are in fact two distinct subpopulations. It is possible that a model that supposes a dynamic decay rate—a qualitatively distinct account of the loss over time—fit the data here as well, if not better. Adjudicating which account is more reliable would likely require not only model comparison but direct experimental testing. The supposition of distinct subpopulation with different change rates predicts that the neurons that decayed rapidly suffered greater damage from the implantation. One possible experiment is to track neurons across days and examine whether neurons that survived longer are those that showed fewer signs of damage in their activity, such as bursting activity, immediately after implantation.

## Performance of explanted probes

Measurements in saline of explanted probes revealed that recording sites that had been lowered into the brain showed only a very small increase in noise relative to unimplanted probes. This difference was much too small to impair high-quality chronic recordings, which we confirmed with multiple cycles of implantation and explantation. By contrast, the part of the probe that remained above the brain surface exhibited local regions with very high noise. This is

presumably related to the application of silicone elastomer and/or petroleum jelly to the exposed part of the shank during surgery, which may not have been fully removed after cleaning with an enzymatic detergent.

First and foremost, these findings support the conclusion that Neuropixels probes, used with the system outlined here, are able to withstand months of implantation in neural tissue followed by explantation, with negligible reduction in performance of the recording sites lowered into the brain. As for the superficial recording sites, it is difficult to know whether the high noise observed on many probes would impair neural data acquisition upon re-implantation, as the corrosive environment of the brain may dissolve any insulating materials applied in a previous surgery. Supporting this possibility, the noise was highly variable on the superficial electrodes of a probe explanted three times, suggesting that these measurements do not reflect a permanent degradation. Nonetheless, we conclude that it may be wise to lower as many recording sites into the brain as possible during surgery, to shield the superficial electrodes from materials applied in surgery that may be difficult to remove. In addition, identifying cleaning reagents that can remove non-biological materials such as silicone, without damaging the Neuropixels probe, would be useful.

Finally, we observed a high degree of correlation in the noise amplitude measured across distant pairs of recording sites that were addressed by the same acquisition channel. This implicates the signal processing circuitry as the major source of noise in the probes, rather than the recording sites themselves. Because this circuitry is not exposed to brain tissue, this bodes well for the long-term performance of the probes, consistent with our empirical findings. Nonetheless, we did observe a slight decrease in the degree of correlation across banks in explanted probes. Taken together with the slight increase in noise amplitude, this points to a modest degradation of the recording sites after long-term implantation.

## Conclusion

Chronic implantation of microelectrodes has been and will likely continue to be an essential technique in neuroscience. However, the Neuropixels probe—a major advance in the electrophysiological toolkit—has not yet reached its full potential due to technical obstacles associated with long-term chronic implantation. The system described here represents a significant step forward in overcoming these obstacles and achieving highly stable, cost-effective, and parallel recordings from chronically implanted Neuropixels probes in freely moving animals. Our characterization of this system may also serve to inform future development of new electrophysiological probes for chronic use as well as improve implantation methods for species and experimental questions not considered here.

# Materials and methods

**Key resources table**

| Reagent type (species) or resource | Designation | Source or reference | Identifiers | Additional information |
|---|---|---|---|---|
| Strain, strain background (species) | Hla(LE)CVF (*Rattus norvegicus*) | Hilltop Lab Animals | Hla(LE)CVF | Long-evans rat |
| Commercial assay or kit | C and B Metabond Quick Adhesive Cement System | Parkell | S380 | |

*Continued on next page*

*Continued*

| Reagent type (species) or resource | Designation | Source or reference | Identifiers | Additional information |
|---|---|---|---|---|
| Chemical compound, drug | Absolute Dentin dual-cure core composite | Parkell | S305 | |
| | Vetbond Tissue Adhesive | 3M | 70200742529 | |
| | DiI Stain | Thermo Fisher Scientific | D282 | 1 mg/mL in isopropyl alcohol |
| | DOWSIL 3–4680 Silicone Gel Kit | DOW | 3–4860 | |
| | Kwiksil | World Precision Instruments | KWIK-SIL | |
| | Black photopolymer resin | Formlabs | RS-F2-GPBK-04 | |
| | Tough photopolymer resin | Formlabs | RS-F2-TOTL-05 | |
| Software, algorithm | SpikeGLX | https://billkarsh.github.io/SpikeGLX/ | | |
| | Kilosort2 | https://github.com/MouseLand/Kilosort2 | Tag: '1.0' (Dec 16, 2019) | |
| | Bcontrol | https://brodylabwiki.princeton.edu/bcontrol/index.php?title=Main_Page | | |
| Other | 3D printed implant assembly | This paper | | https://github.com/Brody-Lab/chronic_neuropixels/tree/master/Holder20CAD20Files |
| | Stereolithography printer | Formlabs | Form two or Form 3 | |
| | Slit corrugated sleeving | McMaster | 2569K93 | |
| | Neuropixel 1.0 Probe with a flat silicon cap | IMEC | PRB_1_4_0480_1 | |
| | Model 1766-AP Cannula Holder | Kopf | 1766-AP | |
| | Silver wire 0.01'/254 μm diameter | A-M Systems | 782500 | |
| | Infusion cannula | Invivo1 | C315GMN/SPC | |
| | Cyanoacrylate accelerator | Pacer Technology | PT-29 | |
| | Conductive copper foil electrical tape | McMaster | 76555A714 | |
| | Low-temperature cautery | Bovie | AA00 | |
| | Tergazyme | Sigma-Aldrich | Z273287-11KG | |
| | PXI Chassis | NI | PXIe-1071 | |
| | PXI Multifunction I/O Module | NI | PXI-6133 | |
| | Noise Rejecting, Shielded BNC Connector Block | NI | BNC-2110 | |
| | PXI Waveform Generator | NI | PXIe-5413 | |
| | RF attenuator | Pomona Electronics | 4108–20 DB | |

## Subjects

A total of 18 male Long-Evans rats (*Rattus norvegicus*) were used in this study. Four of them were BAC transgenic rats expressing Cre recombinase. These rats were used for the purpose of targeting optogenetic constructs to cell types of interest for experiments not described in the present report. These rats came from the following three lines: LE-Tg(Pvalb-iCre)2Ottc (n = 1), LE-Tg(Gad1-iCre) 3Ottc (n = 2), LE-Tg(Drd2-iCre)1Ottc (n = 1). These lines were made by the Transgenic Rat Project at

the National Institute of Drug Abuse (NIDA) and were obtained from the Rat Resource and Research Center (RRRC) at the University of Missouri.

All rats were water restricted to motivate them to work for water as reward, and obtained a minimum volume of water per day equal to 3% of their body mass. If rats consumed less than this minimum during performance of the task, additional water was offered after training. Rats were kept on a reversed 12 hr light-dark cycle and were trained in their dark cycle. Rats were pair housed whenever possible during early training, but were always single housed after implantation to prevent damage to the implant. Starting from the day before each surgery to at least three days after the surgery, rats were given ab lib access to water. Animal use procedures were approved by the Princeton University Institutional Animal Care and Use Committee and carried out in accordance with National Institute of Health standards.

## Implant construction

All parts except the probe itself and the screws were 3D printed in-house using Formlabs SLA printers (Form two and Form 3). CAD files can be found at https://github.com/Brody-Lab/chronic_neuropixels (*Luo and Brody, 2020*; copy archived at https://github.com/elifesciences-publications/chronic_neuropixels). Prints used standard black Formlabs resin (Formlabs; #RS-F2-GPBK-04), except for the headstage mount which was printed in 'Tough' Formlabs resin (Formlabs; #RS-F2-TOTL-05), required for the flexible latching mechanism. After printing, parts were visually inspected for quality and gently sanded down when necessary to ensure correct mating between parts.

After printing and preparing the parts, we slid the internal holder and dovetail adapter onto each other using the dovetail joint (*Figure 2A*) and secured them together using two 1/16" set screws (McMaster; #91375A050; *Figure 2B*). Before mounting the probe, we first secured the internal holder to a stereotaxic cannula holder (Kopf, Tujunga, CA, USA; Model 1766-AP Cannula Holder) held in place by a vise, such that the platform of the dovetail adapter was facing up (*Figure 2C*). The probe was then gently placed on the platform, such that the silicon spacer was lying flat against the platform (*Figure 2D*), and then glued in place with a few drops of thin-viscosity cyanoacrylate glue (Mercury Adhesives) applied using a syringe and 23 gauge needle at the gaps between the probe and dovetail adapter (*Figure 2E*). Care was taken not to apply glue in a way that adheres the dovetail adapter and internal holder, which would prevent removal of the probe from the internal holder for reuse. This was aided by application of cyanoacrylate accelerant (Zip Kicker, Pacer Technology) to quickly harden the glue before it can spread beyond the site of application. We soldered a bare silver wire (0.01" diameter, ~10 cm long) to the bottom-most gold pads on the probe flex cable, shorting the probe's external ground and reference (*Figure 2F*).

Next, we encased the entire assembly in the external chassis. The chassis was composed of two parts, which were added on to either side of the assembly to fully enclose it. The larger piece was added from the underside of the assembly, and screwed in place with four (4) M1.2 × 4 mm screws (*Figure 2G*). The ground wire was led through a small hole in the external chassis so it could be soldered to a second ground wire implanted subdurally during surgery (*Figure 2H*). Then the smaller 'lid-like' piece of the chassis was placed on top of the assembly and the two chassis parts were glued together again using a drop of cyanoacrylate glue near the wider opening of the external chassis, where it is unlikely for the glue to spread to the internal (*Figure 2I*). Extreme care was taken to avoid any glue contacting the probe or internal holder, which would prevent separation from the chassis during explantation.

Next, adhesive copper foil (McMaster; #76555A714) was wrapped around the external chassis for electromagnetic shielding as well as to fully enclose any gaps between the two pieces of the external chassis (*Figure 2J*). The shielding was grounded by cold-soldering it to the silver wire (*Figure 2K*). A thin layer of C and B Metabond (Parkell, Edgewood, NY, USA) was then applied to roughly the lower half of the chassis to enable a better bond between the copper foil and the dental acrylic used to cement the assembly to the animal's head during surgery (*Figure 2K*). Finally, petroleum jelly was applied to the ventral surface of the assembly to seal it from any blood or tissue that could seep through after implantation (*Figure 2L*). Entry of biological fluids into the assembly can create a bond between the internal holder and external chassis, preventing explantation. To apply the petroleum jelly, we melted individual droplets using a low-temperature cautery (Bovie, AA00) and gently dropped them onto the assembly, taking care to avoid contact between cautery's tip and the probe.

A subset of implants (n = 7) were made using an earlier version of the 3D design that embodied a similar design principle but were slightly larger in size. Results were similar between the two versions of the design and were therefore combined.

## Implantation

Surgeries were performed using techniques that were similar to those reported previously (*Erlich et al., 2011*). Rats were placed in an induction chamber under 4% isoflurane for several minutes to induce anesthesia and then injected with 10 mg ketamine and 6 μg buprenorphine intraperitoneally to further assist induction and provide analgesia. Rats were then placed on a stereotaxic frame (Kopf Instruments) and anesthesia was maintained with 1–2% isoflurane flowing continuously through a nose cone. After verifying surgical levels of anesthesia, rats were secured in ear bars, shaved and cleaned with ethanol and betadine. A midline incision was then made with a scalpel, and a spatula was used to clean the skull of all overlying tissue.

A crater-shaped craniotomy roughly 1 mm in diameter was made at the implantation site of each probe and ground (wire, pin, or cannula) (*Figure 3A*). A needle was used to cut an approximately 0.5 mm slit in the dura into which the probe or ground was later lowered.The craniotomy for the ground is distant from the brain region(s) to be recorded, typically in the olfactory bulb or cerebellum. Saline soaked Gelfoam (Pfizer Injectables) was placed in the craniotomies to protect the brain while a thin coat of C and B Metabond (Parkell, Inc) was applied to the skull surface. Then, the Gelfoam was removed from the ground craniotomy and the ground was lowered into the brain.

The ground craniotomy was sealed with silicone elastomer (Kwik-Sil; WPI) and the ground was fixed in place using a small amount of dental composite (Absolute Dentin, Parkell, Inc). Rarely, a second surgery was required to improve the animal ground. For this reason, we subsequently favored a grounding strategy in which a ground wire, pin, or cannula is inserted 1 mm or more into the brain, to ensure a robust electrical connection with brain tissue. A 26-gauge steel ground cannula (*Figure 3C–D*; Invivo1, #C315GMN/SPC) had the advantage of being easy to manipulate using the same stereotaxic arm we used to lower the implant assembly. However, other established methods for creating an animal ground (such as skull screws, which we did not try) may be sufficient if executed properly.

Gelfoam was removed one by one from the remaining craniotomies and the probes inserted. Just before insertion, the probe tip was dipped for several seconds in lipophilic dye (DiI or DiO, 1–2 mg/mL in isopropyl alcohol) to enable histological reconstruction of the probe locations (see examples in *Figure 4—figure supplement 1*). Probes were manually lowered at a rate of 0.1 mm every 10–30 s, and saline was applied to the craniotomy every 30 s to maintain tissue hydration After a probe was fully inserted, a small quantity (<5 μL) of soft silicone elastomer (DOWSIL 3–4680, Dow Chemical) was injected to seal the craniotomy. We found that harder silicones, such as Kwik-Sil (WPI), could damage the probe shank upon application. Other surgical protocols for implanting Neuropixels probes call for completely covering the probe shank with silicone or petroleum jelly to prevent damage to the probe from subsequent application of dental acrylic which quickly hardens and could damage the shank. Instead, we only applied silicone to seal the craniotomy, leaving some of the probe shank exposed to air. Then, we carefully extruded a viscous dental composite (Absolute Dentin, Parkell) through a mixing tip, creating a wall-like barrier connecting the four ventral edges of the external chassis and the skull. This creates a watertight seal around, but not directly contacting, the probe shank (*Figure 3B*). After these steps were completed for each probe, the silver wires soldered to the ground and external reference pads on each probe flex cable were then soldered to the common animal ground (*Figure 3D*). Acrylic resin (DuraLay, Reliance Dental) was applied at low viscosity to fill all gaps and secure the entire implant assembly to the Metabond-coated skull. The ground wires were also covered by the acrylic to prevent them being disturbed after the surgery. Rats were given ketoprofen after the surgery and again 24 and 48 hr post-operative and recovered with ad lib access to water for 5 days before returning to water restriction.

## Explantation

To explant a probe, the animal was first anesthetized and placed in the stereotaxic frame. Then, a stereotaxic cannula holder was attached to the probe's internal holder and the screws fixing the internal holder and external chassis were removed. The stereotaxic arm was raised until the internal

holder, carrying the probe, was fully removed (*Figure 3E*). The internal holder sometimes adhered to the external chassis after screw removal. In these cases, the external chassis was carefully drilled away with a dental drill until the internal holder could be easily removed. After explantation, the probe shank was fully immersed in 1% tergazyme (Alconox) for 24–48 hr, followed by a brief rinse in distilled water and isopropyl alcohol, in that order.

## Electrophysiological recordings

Electrophysiological recordings were performed using either commercially available Neuropixels 1.0 acquisition hardware (*Putzeys et al., 2019*) or earlier test-phase IMEC acquisition hardware. The former was used in conjunction with PCI eXtensions for Instrumentation (PXI) hardware from National Instruments (including a PXIe-1071 chassis and PXI-6133 I/O module for recording analog and digital inputs.) We used SpikeGLX software (*Karsh, 2020*) to acquire the data. For measurement of signal stability over time, the selected reference was a silver wire shorted to the ground wire and penetrating the brain at a different location from the probe insertion site. The amplifier gain used during recording was 500. The recording channels addressed either the deepest 384 recording sites ('bank 0') or the second deepest 384 recording sites ('bank 1'), and recordings lasted approximately ten minutes. Recordings were made on days with logarithmically spaced intervals.

Spikes were sorted offline using Kilosort2 (*Pachitariu, 2020*), using default parameters and without manual curation. A unit was considered a single-unit according to the default Kilosort2 criteria, as summarized here. A putative single unit must meet both of the following conditions: (1) the proportion of refractory violations must be less than 5%; and (2) the number of refractory violations must not be trivially predictable from Poisson statistics ($p \leq 0.05$). The latter condition prevented low-firing rate multi-units from being counted as single units simply because too few spikes were observed to estimate the proportion of refractory period violations. To determine if a unit met these conditions, the autocorrelogram was computed in 1 ms bins and the number of refractory period violations was computed as the sum of the values in the central bins. The proportion in condition one was computed by comparing the number of refractory violations with a baseline sum calculated from the shoulder of the autocorrelogram. The span of the central bins varied from +/- 1.5 ms to +/- 10.5 ms, and the span of the shoulder bins were either +/- 10.5 ms to +/- 50 ms or +/- 25.5 ms to +/- 499.5 ms. A unit was considered a single unit if it met criterion 1 for at any combination of the central and shoulder spans and also criterion two for any of the central spans.

## Sum-of-exponentials model

In *Figure 4D–L* and *Figure 4—figure supplement 6*, the number of units recorded from each session was modelled as a sum of two exponential decay terms:

$$N = N_1 \left( \alpha e^{k_{fast}(t-1)} + (1-\alpha)e^{k_{slow}(t-1)} \right) + \varepsilon \tag{1}$$

$N$ is the number of units (or SUs), $N_1$ is the number units on the first day after implant, $t$ is the number of days after the implantation, $\alpha$ indicates the fraction of the units that are rapidly decaying with the change rate $k_{fast}$, and the remaining fraction $(1-\alpha)$ decays with a slower (less negative) change rate $k_{slow}$, and, $\varepsilon \sim N(0, \sigma^2)$. The estimated parameters were found by maximizing the log-likelihood using MATLAB's *fmincon* function, using the interior-point algorithm, and with the constraints that $-10 < k_{fast} \leq k_{slow} < -10^{-6}$. The exponential terms at these bounds are effectively 0 or 1, and the bounds were imposed to improve the consistency of the fitting. Error in the coefficient estimates was estimated as using the percentile bootstrap confidence intervals computed from 1000 bootstrap samples.

In *Figure 5*, the sum-of-exponentials regression (SoER) model was fit to the number of units recorded from each electrode from each recording session:

$$N = Poisson\left(\lambda\right) \tag{2}$$

$$\lambda = N_1 \left[ \alpha e^{k_{fast}(t-1)} + (1-\alpha)e^{k_{slow}(t-1)} \right] \tag{3}$$

$$N_1 = \beta_0^{N_1} + \beta_{AP}^{N_1} \cdot AP + \beta_{DV}^{N_1} \cdot DV + \\ \beta_{DV>-2}^{N_1} \cdot I_{DV>-2} + \beta_{ML}^{N_1} \cdot ML \tag{4}$$

$$k_{fast} = \beta_{fast} + k \tag{5}$$

$$k_{slow} = \beta_{slow} + k \tag{6}$$

$$k = \beta_{AP}^{k} \cdot AP + \beta_{DV}^{k} \cdot DV + \beta_{DV>-2}^{k} \cdot I_{DV>-2} + \\ \beta_{ML}^{k} \cdot ML + \beta_{SP}^{k} \cdot SP + \beta_{SO}^{k} \cdot SO \tag{7}$$

The number of units recorded on an individual electrode during a session ($N$) depends on the unit count on the first day after implant ($N_1$), which is a weighted sum of the regressors $AP$ (mm anterior), $DV$ (mm brain surface), $I_{DV>-2}$ (one if the electrode has a $DV$ position greater than $-2$ mm, which is approximately dorsal cerebral cortex, and 0 otherwise), and $ML$ (mm from midline), plus a constant term. In variants of the model that are not shown, we also included regressors that were categorical variables along the AP or ML axis, but those regressors were consistently excluded by the parameter selection process (see below). The model supposes two distinct subpopulations with different rates of exponential change. A fraction of units ($\alpha$) changes exponentially at the rate of $k_{fast}$, and the remaining fraction of units ($1-\alpha$) changes at the rate of $k_{slow}$. The $k$ term within the $k_{fast}$ and $k_{slow}$ terms is the same, and the β coefficients within $k$ are shared by $k_{fast}$ and $k_{slow}$. The terms $k_{fast}$ and $k_{slow}$ differ only by a constant offset $\beta_{fast}$- $\beta_{slow}$. The change rates of both subpopulations depend on a weighted sum of the regressors $AP$, $DV$, $I_{DV>-2}$, $ML$, $SP$ (mm from the tip of the probe shank), and $SO$ (shank orientation, angle between the shank's plane and the brain's sagittal plane), as well as a constant term.

Because the regressors are partly correlated, the estimate of the coefficient of each regressor, as well as the error of the estimate, depend on what other regressors are included in the model. To make the inference more reliable, we first selected the subset of regressors that allowed the model to best predict unit count over time. During all-subset variable selection, model variants included either the continuous regressor $DV$ or the categorical regressor $I_{DV>-2}$, but not both. The number of model variants were 448 ($2^{10}$–576 subsets with both the $DV$ and $I_{DV>-2}$ regressors in either the $N_1$ or $k$ term). In addition to the regressor coefficients, all model variants included the variables $\alpha$, $\beta_{fast}$, $\beta_{slow}$, a constant in $N_1$.

The SoER model was fitted by maximizing log-likelihood using MATLAB's *fmincon* function, with the constraint that $-10 < k_{fast} \leq k_{slow} < -10^{-6}$. All subsets of the variables were fit in different model variants and evaluated using out-of-sample log-likelihood (LL) computed using five-fold cross-validation. To avoid underestimating the confidence intervals, the data was randomly divided in halves, separately used for variable selection and for estimating the error of coefficient estimates. The procedure involves randomly splitting the data, selecting the best model using one half of the data, and using the other half to estimate the coefficients, and the process was repeated ten times to confirm that the results were consistent. The point and error estimates of the coefficients shown in *Figure 5* were from the repetition in which the best model's out-of-sample LL was the median across the 10 repetitions. Error in the coefficient estimates were 95% percentile bootstrap confidence intervals calculated from 1000 bootstrap samples.

Conclusions were not changed if a Gaussian-noise model were used. A single-term exponential model was associated with much lower out-of-sample log-likelihood and therefore not used. The all-subset variable selection procedure was preferred over stepwise selection due to its greater consistency in model selection, and it was used instead of LASSO because it allows comparison between the best model and the runner-up models.

To confirm the results of the SoER model, an elaborated SoER model was fitted (*Figure 5—figure supplement 1*). Two additional regressors were introduced: *age*, the animal's age in days, and *use*, the number of times a probe had been previously used for implantation (0–2). Also, the fast-decaying and slow-decaying population depends on separate linear combinations of the regressors (rather than a single linear combination and a scaling parameter). Due to the larger number of parameters

(21), feature selection was performed using L1 regularization rather than all-subset selection (i.e. L0 regularization).

$$N = Poisson\left(\lambda\right) \tag{8}$$

$$\lambda = Ae^{k_{fast}(t-1)} + Be^{k_{slow}(t-1)} \tag{9}$$

$$A = \beta_0^A + \beta_{AP}^A \cdot AP + \beta_{DV>-2}^A \cdot I_{DV>-2} + \\ \beta_{ML}^A \cdot ML + \beta_{age}^A \cdot age + \beta_{use}^A \cdot use \tag{10}$$

$$B = \beta_0^B + \beta_{AP}^B \cdot AP + \beta_{DV>-2}^B \cdot I_{DV>-2} + \\ \beta_{ML}^B \cdot ML + \beta_{age}^B \cdot age + \beta_{use}^B \cdot use \tag{11}$$

$$k_{fast} = \beta_{fast} + k \tag{12}$$

$$k_{slow} = \beta_{slow} + k \tag{13}$$

$$k = \beta_{AP}^k \cdot AP + \beta_{DV>-2}^k \cdot I_{DV>-2} + \\ \beta_{ML}^k \cdot ML + \beta_{age}^k \cdot age + \beta_{use}^k \cdot use + \\ \beta_{SP}^k \cdot SP + \beta_{SO}^k \cdot SO \tag{14}$$

The coefficient estimates were found by minimizing the sum of the negative Posson log-likelihood and a penalization term based on the L1 norm of the parameters. The constant parameters $\beta_0^A$, $\beta_0^B$, $\beta_{fast}$, $\beta_{slow}$ were not penalized, and in the penalization term, the coefficients $\beta^k$ were multiplied by a weight parameter so that they contribute similarly as the coefficients $\beta_0^A$ and $\beta_0^B$. The weight parameter was computed using the coefficient estimates when the model was fitted without regularization. The procedure to estimate the coefficients splits the data randomly in half and uses one half was used to estimate the optimal scaling of the penalization term. This optimal penalization scaling parameter and the other half of the data were then used to estimate the parameters.

## Behavioral task

Training took place in an operant box with three nose ports (left, center and right) and two speakers, placed above the right and left nose ports. Each trial began with a visible light-emitting diode (LED) turning on in the center port. This cued the rat to insert its nose into the center port and keep it there until the LED was turned off (1.5 s 'fixation' period). In each trial, rats were concurrently presented with two trains of auditory pulses, one train from a loudspeaker on its left, and the other from a loudspeaker on its right (*Brunton et al., 2013*). At the end of each trial, the subjects received a reward if they oriented to the port on the side of the auditory train with the greater total number of pulses. The timing of pulses varied both within and across individual trials and between the left and right streams. Behavioral control and stimulus generation used Bcontrol software (brodylabwiki. princeton.edu/bcontrol). Before probe implantation, subjects were trained in a semi-automated, high-throughput training facility, each in a behavioral enclosure 13' wide, 11' deep and 18.5' tall within a sound-attenuated chamber (Coulbourn, Holliston, MA, USA). After surgery, tethered behavioral performance took place in a larger behavioral enclosure that was 13' wide, 11' deep and 40' tall within a sound-attenuated chamber and built-in Faraday cage (IAC, Naperville, IL, USA). Untethered performance took place in the training facility, as before surgery.

## Behavioral performance metrics

The performance of each rat (pooled across multiple sessions) was fit with a four-parameter logistic function:

$$P(right) = \gamma 0 + \gamma 1/(1 + exp(-\beta(x - \alpha))) \tag{15}$$

where *x* is the click difference on each trial (number of right clicks minus number of left clicks), and *P (right)* is the fraction of trials when the animal chose right. The parameter $\alpha$ is the x value (click difference) at the inflection point of the sigmoid, quantifying the animal's bias; $\beta$ quantifies the sensitivity of the animal's choice to the stimulus; $\gamma_0$ is the minimum *P(right)*; and $\gamma_0+\gamma_1$ is the maximum *P (right)*. The lapse rate is $(1-\gamma_1)/2$. The number of trials completed excludes trials when the animal prematurely broke fixation and trials after the animal in which the animal failed to choose a side port after 5 s.

### Measurement of input-referred noise

The RMS noise of the AP band (300 Hz - 10 kHz) of each recording site was measured in a 0.9% phosphate buffered saline (PBS) solution, following the procedure described in the Neuropixels User Manual (IMEC). To accurately compare across recording sites and probes, this procedure requires first determining a gain correction factor for each recording site, so that the nominal and actual gains match. To do this, we measured the response to a 4mVpp, 1.8 kHz sine wave generated using an arbitrary waveform generator (PXIe-5413, National Instruments) in 0.9% PBS. A 400 mVpp waveform was generated by the PXIe-5413 and then attenuated 100 fold using two stacked 20 dB attenuators (Digikey, 501–1527-ND). We compared the measured amplitudes on each recording site to the amplitude measured on an independent, calibrated I/O module (PXI-6133, National Instruments) to determine the gain correction factor.

### Choice selectivity

This metric is based on the receiver operating characteristic (ROC) and indexes how well an ideal observer can classify left- versus right-choice trials using the spike counts of an isolated unit. Spikes were counted in 0.1 s bins stepped in 0.02 s, from 1 s before movement onset to 0.5 s afterwards. Trials were included for analysis if the rat did not prematurely disengage from the center port and also reported its choice within 5 s after it was cued to do so. An ROC curve classifying left- and right-choice trials was constructed based on the spike counts of each unit in each time bin. The area under the ROC curve ranged from 0 to 1, with values greater than 0.5 indicating a larger mean spike count associated with right-choice trials. Because the present analysis concerns only the magnitude and not the directionality of the choice selectivity, a value *x* less than 0.5 was flipped to the corresponding value above 0.5, that is |x-0.5| + 0.5. The choice selectivity results were from the first recording session for each animal after the implantation when the animal had completed more than a hundred trials (4–11 days after surgery).

## Acknowledgements

We thank T Harris, M Oostland, and M Schottdorf for comments and discussion; J Teran for technical assistance; J Putzeys for advice on measurement of input-referred noise; and J Colonell and B Karsh for advice on the recording equipment. This work was funded by NIH grants R01MH108358 and F32MH115416 and by the Howard Hughes Medical Institute.

## Additional information

### Funding

| Funder | Grant reference number | Author |
|---|---|---|
| National Institute of Mental Health | R01MH108358 | Thomas Zhihao Luo<br>Adrian Gopnik Bondy<br>Diksha Gupta<br>Verity Alexander Elliott<br>Charles D Kopec<br>Carlos D Brody |
| National Institute of Mental Health | F32MH115416 | Thomas Zhihao Luo |
| Howard Hughes Medical Institute | | Thomas Zhihao Luo<br>Adrian Gopnik Bondy<br>Diksha Gupta |

Verity Alexander Elliott
Charles D Kopec
Carlos D Brody

The funders had no role in study design, data collection and interpretation, or the decision to submit the work for publication.

## Author contributions

Thomas Zhihao Luo, Conceptualization, Resources, Data curation, Software, Formal analysis, Supervision, Funding acquisition, Validation, Investigation, Visualization, Methodology, Writing - original draft, Project administration, Writing - review and editing; Adrian Gopnik Bondy, Conceptualization, Resources, Data curation, Software, Formal analysis, Supervision, Validation, Investigation, Visualization, Methodology, Writing - original draft, Project administration, Writing - review and editing; Diksha Gupta, Methodology, Writing - review and editing; Verity Alexander Elliott, Data curation, Investigation, Writing - review and editing; Charles D Kopec, Resources, Data curation, Investigation, Writing - review and editing; Carlos D Brody, Conceptualization, Resources, Supervision, Funding acquisition, Project administration, Writing - review and editing

## Author ORCIDs

Thomas Zhihao Luo (ID) https://orcid.org/0000-0002-7774-1697
Adrian Gopnik Bondy (ID) https://orcid.org/0000-0002-7265-5810
Diksha Gupta (ID) http://orcid.org/0000-0002-8811-3311
Carlos D Brody (ID) http://orcid.org/0000-0002-4201-561X

## Ethics

Animal experimentation: This study was performed in strict accordance with the recommendations in the Guide for the Care and Use of Laboratory Animals of the National Institutes of Health. All of the animals were handled according to approved institutional animal care and use committee (IACUC) protocols (#1853) of Princeton University. All surgery was performed under isofluorane anesthesia, and every effort was made to minimize suffering.

## Decision letter and Author response

Decision letter https://doi.org/10.7554/eLife.59716.sa1
Author response https://doi.org/10.7554/eLife.59716.sa2

# Additional files

## Supplementary files

• Transparent reporting form

## Data availability

All data generated or analyzed during this study can be found at https://doi.org/10.5061/dryad.m63xsj3zw. All code used in preparation of this article can be found at https://github.com/Brody-Lab/chronic_neuropixels copy archived at https://github.com/elifesciences-publications/chronic_neuropixels.

The following dataset was generated:

| Author(s) | Year | Dataset title | Dataset URL | Database and Identifier |
|---|---|---|---|---|
| Luo, Thomas | 2020 | An approach for long-term, multi-probe Neuropixels recordings in unrestrained rats | https://doi.org/10.5061/dryad.m63xsj3zw | Dryad Digital Repository, 10.5061/dryad.m63xsj3zw |

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
