## [Decision Letter]

**Acceptance summary:**

The new techniques presented here for obtaining chronic recordings using multiple neuropixels probes in rats will be of high value to the neuroscience community. The work builds and extends existing work in several important ways – addressing the long-term yield (i.e. number of neurons) across different brain regions, the feasibility of probe re-use, the probe performance over a long period of time and impact of the probe on behavior. Due to the importance of using freely moving animals in neuroscience research, and the differences between rats and mice that necessitate modifications on existing technology, this paper is timely and likely to be very useful to a sizable group of researchers.

**Decision letter after peer review:**

Thank you for submitting your article "An approach for long-term, multi-probe Neuropixels recordings in unrestrained rats" for consideration by *eLife*. Your article has been reviewed by three peer reviewers, including Lisa Giocomo as the Reviewing Editors and Reviewer #1, and the evaluation has been overseen by Laura Colgin as the Senior Editor. The following individual involved in review of your submission has agreed to reveal their identity: Anne K Churchland (Reviewer #2).

The reviewers have discussed the reviews with one another and the Reviewing Editor has drafted this decision to help you prepare a revised submission.

Summary:

This manuscript presents new techniques for obtaining chronic recordings using multiple neuropixel probes in rats. The study provides very informative trends regarding long-term (~4-month) recording with Neuropixels probes in chronically implanted, freely moving rats. This is accomplished by recording across many animals (n = 18) and many recording locations and analyzing the number of single (and multi) units that can be automatically isolated as a function of time since implant, recording location, and other features (e.g. shank orientation). The authors perform these experiments with a modular system that allows the implanting of multiple probes simultaneously in a single rat (here they mostly implanted 1 probe, sometimes 2, once 3) and that allows the removal of probes for re-use in another animal, both of which are also valuable contributions. The work builds on existing technology in important ways: the authors examined the long-term yield across different brain regions, they more extensively assessed the feasibility of probe reuse compared to previous work, and they evaluated probe performance over a long period of time and also after explanation (measuring the input referred noise of explanted probes in saline). Because of the importance of using freely moving animals in neuroscience research, and the differences between rats and mice that necessitate modifications on existing technology, this paper is timely and likely to be very useful to a sizeable group of researchers.

Revisions:

1) The manuscript would fit better into the category of "Tools and Resources" if it included more methodological detail about the construction and implant of the device. For instance, the material from the "Neuropixels implant procedure" is helpful and should be in the Results or Materials and methods. The authors should also use consistent nomenclature across documents (i.e. "chassis" in the main text is referred to as the "external" on the google doc, the part referred to as the "internal" in the google doc is called an "internal holder" in the manuscript). A reader hoping to use the device might also benefit from more information on the grounding procedure. Where on the probe should the ground wire be connected and how one should fix the grounding wire? How should one should protect the grounding wire from being touched by the animal? Are the craniotomy and durotomy necessary for grounding? Could one simply connect the grounding wire to a couple of screws on the skull? Figure 6 in the google doc protocol is really helpful and could instead be in the main manuscript. An additional figure showing how to connect the wire to the ground during the surgery would also be quite useful.

2) The results in Figure 2, especially the averages in Figure 2H,K, indicate severe losses in unit yield over time for probes implanted posterior of bregma and electrode sites in the dorsal 2mm of the rat brain. However, Figure 2B shows at least one animal (open circles) for which high neuron yield was obtained in motor cortex and dorsal striatum for at least 4 months. First, is this from 1 or 2 probes? Whether it is from 1 or 2 probes, the stability of the recording over time from day 1 is much greater than the other animals, and much better than what is expected from Figure 2H. Is the preservation of units over time for this animal due to the stability of units in dorsal striatum (presumably mostly >2mm below the surface) or also motor cortex?

3) The age of the animals could also account for yield differences. The authors should list age at implant in their table in Figure 4—figure supplement 1. The authors should also try adding age as one of the regressors for their model of neuron yield. Another potential cause is whether the probe is new or reused. The authors showed that probe re-use did not result in statistically different yield for the medial prefrontal cortex. But is this also true for the other brain regions? Does the data in Figure 2 include implants of both new and re-used probes, or only new probes? The authors should try to add whether the probe was new or re-used as a regressor in their neuron yield model. Regarding points of whether or not it is possible to add age or probe newness as regressors in the model, the authors should create a supplementary figure that shows the single unit yield curves as in Figure 2A-C for all probes in all animals: one panel per major brain region (e.g. splitting motor cortex from dorsal striatum from ventral striatum), with one curve per probe. There should be a legend for each panel that gives the (AP,ML,DV) coordinates of the approximate midpoint of the probe's location within that brain region. The legend should also indicate for each probe/curve: the animal, age at time of implant, probe newness, probe tip depth, estimated number of electrodes recorded from in that region, and shank orientation. This will repeat some pieces of information that's in the tables, however it's very useful to see all this information together in a form that would be very valuable for readers, especially experimenters who may want to record from some of the more posterior and dorsal areas. The information that could be gleaned would include knowledge of the variance in yield over time across implant attempts, so they could see if, say, 1 of 3 attempts to implant in a given area may give very good long-term yield.

4) It is stated that "The relative number of units corresponding to the fast- and slowly-decaying subpopulations did not significantly vary across brain regions along either anatomical axis, nor did the rate of decay of the fast population (Figure 2—figure supplement 3). This suggests that the rapid decline in yield observed in the days after surgery may be due to a process that is relatively uniform across brain regions." The support for this statement can be seen in the indicated Figure 2—figure supplement 3. On the other hand, the point is made (and shown in Figure 2—figure supplement 4) that there is no loss of units in mPFC over time. This is apparently at odds with the statement and model assumption of a fixed fraction of fast-decaying units. Was a model tried in which α varies with location? If the statement is ultimately kept, there should at least be a comment made there that the most anterior, ventral regions appear to differ from the model's assumption/interpretation.

5) Did the authors consider comparing their Kilosort2 unit identification with manual curation or another sorting software?

6) Figure 2 is perhaps one of the most informative findings but I wonder how applicable this will be to future probe iterations. Do the authors have a hypothesis for what features of the probe might contribute (or not contribute) to the long term loss of units?

---

## [Author Response]

Revisions:1) The manuscript would fit better into the category of "Tools and Resources" if it included more methodological detail about the construction and implant of the device. For instance, the material from the "Neuropixels implant procedure" is helpful and should be in the Results or Materials and methods. The authors should also use consistent nomenclature across documents (i.e. "chassis" in the main text is referred to as the "external" on the google doc, the part referred to as the "internal" in the google doc is called an "internal holder" in the manuscript). A reader hoping to use the device might also benefit from more information on the grounding procedure. Where on the probe should the ground wire be connected and how one should fix the grounding wire? How should one should protect the grounding wire from being touched by the animal? Are the craniotomy and durotomy necessary for grounding? Could one simply connect the grounding wire to a couple of screws on the skull? Figure 6 in the google doc protocol is really helpful and could instead be in the main manuscript. An additional figure showing how to connect the wire to the ground during the surgery would also be quite useful.

We thank the reviewers for this feedback, which has dramatically improved the manuscript. We have increased greatly the amount of information detailing the construction and implantation procedures in the main text, including adding two new figures. In the revised manuscript, Figure 2 contains images demonstrating the implant construction procedure, including the specific steps that are suggested by the reviewer. Figure 3 provides images demonstrating the key surgical procedures associated with implantation and explantation. In addition, the Materials and methods section has been significantly expanded to contain additional detailed information about implant construction, implantation and explantation. Finally, we plan to separately submit a protocol associated with the manuscript to BioProtocols (or a similar protocol database) which will contain even more detailed information, including a full parts list. We have also taken care to be fully consistent in our use of terminology throughout all components of the manuscript, as rightly suggested by the reviewer.

With regards to grounding, the reader will find answers to all the specific questions raised by the reviewer in the newly added material, including pictures specifically illustrating the grounding methods during implant construction and surgery (see new Figure panels 2F and 3C,D). We briefly summarize our grounding procedures here. First a 10cm silver ground wire was soldered onto the most ventral pads on the probe flex cable (i.e. those closest to the probe base), and then it emerged through a dedicated hole in the external chassis. The shield was grounded by cold-soldering it to the ground wire. To create an animal ground, we used a variety of methods as described in the revised manuscript. The method we used most often, and for which we now provide pictures, involves directly implanting a 26-gauge steel cannula, soldered onto a 5cm length of silver wire, 1mm into the brain. We reasoned that implanting a large amount of conductive material directly in the brain in this fashion provided the best chance of a robust animal ground, particularly after the early experience of several poorly grounded probes when other methods were used. We have no data to determine whether other methods that we did not try, such as skull screws, may also be sufficient.

2) The results in Figure 2, especially the averages in Figure 2H,K, indicate severe losses in unit yield over time for probes implanted posterior of bregma and electrode sites in the dorsal 2mm of the rat brain. However, Figure 2B shows at least one animal (open circles) for which high neuron yield was obtained in motor cortex and dorsal striatum for at least 4 months. First, is this from 1 or 2 probes? Whether it is from 1 or 2 probes, the stability of the recording over time from day 1 is much greater than the other animals, and much better than what is expected from Figure 2H. Is the preservation of units over time for this animal due to the stability of units in dorsal striatum (presumably mostly >2mm below the surface) or also motor cortex?

Each set of markers in Figure 2B corresponds to data from a single probe. In Figure 2B, the probe indicated by the open circles shows a stability in single unit yield that appears to be much better than what is expected from Figure 2H because it combines data from dorsal striatum and motor cortex. The plot in Author response image 1 shows the yield from the same probe, separating the data from motor cortex and dorsal striatum. It indicates that the stability in single unit count in this probe is mostly accounted for by dorsal striatum.

**Author response image 1. sa2fig1:** 

Because motor cortex and dorsal striatum have such different time courses in unit yield, Figure 2B has been modified to include only single units from motor cortex.Figure 2B in the original manuscript is now Figure 4B.

3) The age of the animals could also account for yield differences. The authors should list age at implant in their table in Figure 4—figure supplement 1. The authors should also try adding age as one of the regressors for their model of neuron yield. Another potential cause is whether the probe is new or reused. The authors showed that probe re-use did not result in statistically different yield for the medial prefrontal cortex. But is this also true for the other brain regions? Does the data in Figure 2 include implants of both new and re-used probes, or only new probes? The authors should try to add whether the probe was new or re-used as a regressor in their neuron yield model. Regarding points of whether or not it is possible to add age or probe newness as regressors in the model, the authors should create a supplementary figure that shows the single unit yield curves as in Figure 2A-C for all probes in all animals: one panel per major brain region (e.g. splitting motor cortex from dorsal striatum from ventral striatum), with one curve per probe. There should be a legend for each panel that gives the (AP,ML,DV) coordinates of the approximate midpoint of the probe's location within that brain region. The legend should also indicate for each probe/curve: the animal, age at time of implant, probe newness, probe tip depth, estimated number of electrodes recorded from in that region, and shank orientation. This will repeat some pieces of information that's in the tables, however it's very useful to see all this information together in a form that would be very valuable for readers, especially experimenters who may want to record from some of the more posterior and dorsal areas. The information that could be gleaned would include knowledge of the variance in yield over time across implant attempts, so they could see if, say, 1 of 3 attempts to implant in a given area may give very good long-term yield.

We have included the animal’s age on the day of the implantation in the table that provides details of each probe insertion (now Table 1; previously an unnamed table in the Materials and methods). An additional table was created (Table 2) that details each brain region recorded by each implant, with information on the animal’s age, the (AP,ML,DV) coordinates of the approximate midpoint of the probe's location within that brain region, the animal, age at time of implant, probe newness, probe tip depth, estimated number of electrodes recorded from in that region, and shank orientation.

A new supplemental figure (Figure 4—figure supplement 3) plots the single unit yield curve for each probe, grouped by brain region. Each curve is labelled by its implant number, which can be looked up in Tables 1 and 2 for information on each probe and the number of electrodes of that probe in a brain region.

The only brain region that was targeted with previously used probes was medial frontal cortex. Nonetheless, we explored the effect of probe re-use on initial unit count and change rate using the regression model fitted to unit count on each electrode. We included in the model parameters that were previously found to be predictive of the unit count over time (α,βfast,βslow,β0N1,βAPN1,βM LN1,βAPκ,βDV>2κ,βSOκ,) , and we introduced the parameters to assess the rat’s age at the time of implant (238-650 days) and probe’s number of previous use (0-2) on either the initial yield (βageN1,βuseN1) or exponential change rate (βagek,βusek). All-subset variable selection was performed to exhaustively identify the subset of parameters (among 512 subsets) that best predicted the data among the parameters (βAPN1,βM LN1,βageN1,βuse,N1βAPκ,βDV>2κ,βSOκ,βagek,βusek). The results of the model fitting confirmed the relevance of the parameters that were previously found to be predictive. They also showed that age affected the initial unit count (older animals had fewer initial yields) and the change rate (older animals showed slower decay). The results also indicated that previous use is correlated with modestly faster exponential decay, but this finding is based on only two implants with used probes.

Similar results were found using an elaborated sum-of-exponential regression model (related to comment #4) that includes even the parameters that were previously rejected to be relevant and also allows the initial counts of the fast-decaying and slow-decaying populations to separately depend on the regressors. The results of this model are presented in Figure 5—figure supplement 1.

4) It is stated that "The relative number of units corresponding to the fast- and slowly-decaying subpopulations did not significantly vary across brain regions along either anatomical axis, nor did the rate of decay of the fast population (Figure 2—figure supplement 3). This suggests that the rapid decline in yield observed in the days after surgery may be due to a process that is relatively uniform across brain regions." The support for this statement can be seen in the indicated Figure 2—figure supplement 3. On the other hand, the point is made (and shown in Figure 2—figure supplement 4) that there is no loss of units in mPFC over time. This is apparently at odds with the statement and model assumption of a fixed fraction of fast-decaying units. Was a model tried in which α varies with location? If the statement is ultimately kept, there should at least be a comment made there that the most anterior, ventral regions appear to differ from the model's assumption/interpretation.

We have implemented an elaborated sum-of-exponentials regression model in which initial counts of fast-decaying and slow-decaying units no longer have a constant ratio but instead depend on separate linear combinations of the regressors. The terms *N* and 𝛼 were replaced with two terms *A* and *B*, representing the initial unit count of the fast-decaying and slow-decaying population, respectively. The out-of-sample log-likelihood of this model is very modestly but reliably higher than the original model (0.6%; *p < 0.001*). The terms *A* and *B* had different estimates for the coefficients of the regressors *AP* and *ML*, thereby providing evidence that the relative proportion of fast-decaying units did depend on the brain regions. Therefore, we have removed the statement in and added a description of the elaborated model, which is provided in Figure 5—figure supplement 1. The elaborated model is presented as a supplemental figure because of its complexity and because the results of the elaborated model corroborates the results in the original, simpler model, namely, that the initial unit counts depended on the regressors *AP* and *ML*, and the change rates depended on the regressors *AP*, *DV>-2* (dorsal cortex), and *SO* (shank orientation).

5) Did the authors consider comparing their Kilosort2 unit identification with manual curation or another sorting software?

Our spike sorting pipeline typically does include manual curation by a trained technician. However, for the purposes of this manuscript, we chose to report the uncurated data. This measure has the advantage of being entirely reproducible (because we used an automatic sorter and default parameters) and immune from the subjectivity and variability of curation, at the expense of some degree of unit quality. Because the primary purpose of the present report is to compare yield across sessions while neural responses were not of primary interest, these tradeoffs seemed appropriate.

However, in light of the reviewer’s comment, we have performed additional analyses to compare the uncurated and curated data for those subset of sessions (n=25) in which curation had been performed. These analyses are included as a supplementary figure (Figure 4—figure supplement 2) in the revised submission. The primary effect of manual curation was to uniformly scale down the number of identified units per session, consistent with a known high false positive rate of Kilosort2 (Buccino et al., 2020), with little impact on the relative yields across sessions (correlation of number of SUs in the pre- and post-curation data was 0.97 across sessions). As a result, manual curation has only a minimal impact on relative measures like the dynamics of yields across the lifespan of an implant.

As for the choice of spike sorting software, we chose Kilosort2 because it has a wide adoption, compares favorably to other sorters designed for high-yield silicon arrays (Buccino et al., 2020), and is designed to track units despite drift of the probe relative to the brain. For this paper, we did not compare the output of Kilosort2 to other sorters. Existing data on the subject show strong differences in the number of units detected per session across spike sorters, but it is not well understood how or whether these yield differences depend on features of the data. We speculate that relative measures, such as the dynamics of yield across sessions over the lifespan of an implant statically fixed in a single brain area, would be only minimally affected by the choice of sorter, as we observed in our comparison between curated and uncurated data. However, we cannot rule out the possibility that some of our results would come out slightly differently if other spike sorting software had been used, as we now explicitly mention.

We discuss the points above in a new paragraph in the “Stability of spiking signals” section of the Results.

6) Figure 2 is perhaps one of the most informative findings but I wonder how applicable this will be to future probe iterations. Do the authors have a hypothesis for what features of the probe might contribute (or not contribute) to the long term loss of units?

This is a useful question to consider further, and we have expanded on these points in two new paragraphs in the Discussion (last two paragraphs of the “Long-term stability of neural signals” section). In short, we conclude that a smaller cross-sectional area and improved mechanical flexibility may contribute to better long-term yields in future probe iterations.